# Podocyte specific knockout of the natriuretic peptide clearance receptor is podocyte protective in focal segmental glomerulosclerosis

Liming Wang[1], Yuping Tang[1], Anne F. Buckley[2], Robert F. Spurney[1]*

1 Division of Nephrology, Department of Medicine, Duke University and Durham VA Medical Centers, Durham, North Carolina, United States of America, 2 Department of Pathology, Duke University Medical Center, Durham, North Carolina, United States of America

* robert.spurney@duke.edu

## Abstract

Natriuretic peptides (NPs) bind to glomerular podocytes and attenuate glomerular injury. The beneficial effects of NPs are negatively regulated by the NP clearance receptor (NPRC), which is highly expressed in podocytes. To determine if inhibiting NPRC is podocyte protective, we examined the effects of deleting NPRC in both cultured podocytes and in vivo. We found that: 1.Both atrial NP and C-type NP inhibit podocyte apoptosis in cultured podocytes, but these podocyte protective effects are significantly attenuated in cells expressing NPRC, and 2. Atrial NP was significantly more effective than CNP at inhibiting the apoptotic response. Consistent with the protective actions of NPs, podocyte specific knockout of NPRC reduced albuminuria, glomerular sclerosis and tubulointerstitial inflammation in a mouse model of focal segmental glomerulosclerosis. These beneficial actions were associated with: 1. Decreased expression of the myofibroblast marker alpha-smooth muscle actin, 2. Reduced expression of the extracellular matrix proteins collagen 4-alpha-1 and fibronectin, and 3. Preserved expression of the podocyte proteins nephrin and podocin. Inhibiting NP clearance may be a useful therapeutic approach to treat glomerular diseases.

## Introduction

Focal segmental glomerulosclerosis (FSGS) is an important cause of kidney disease worldwide [1]. The disease is characterized by heavy proteinuria, scarring of the glomerulus in a segmental fashion and frequent progression to end stage kidney disease (ESKD [2]. Treatment is directed at controlling hypertension and reducing proteinuria by blockade of the renin-angiotensin system [3]. More recently, sodium glucose transporter 2 inhibitors (SGLT2i) have further slowed disease progression [4–6]. Despite these advances, nephrotic patients with FSGS remain at significant risk for the development of ESKD [3].

Glomerular podocytes are terminally differentiated cells that play a key role in maintaining both the structure of the filtering apparatus and glomerular permselectivity [2,7,8]. While

**Data availability statement:** All relevant data are within the manuscript and its Supporting Information files.

**Funding:** 1. R21 TR004257-01 from the National Institutes of Health. 2. BX002984 from the Veterans Administration Merit Review Program. 3. W81XWH-19-1-0314 from the United States Department of Defense. The funders had no role in study design, data collection and analysis, decision to publish, or preparation of the manuscript.

**Competing interests:** NO authors have competing interests.

multiple mechanisms cause FSGS, the unifying feature is podocyte injury that leads to a reduction in the number of viable podocytes [2,7]. Because podocytes are unable to proliferate, the decrease in glomerular podocytes promotes damage to the filtering apparatus, which eventually causes glomerulosclerosis (GS), and renal failure.

NPs bind to cell surface NP receptors (NPRs) [9–11]. Atrial NP (ANP) and brain NP (BNP) bind NPR type A (NPRA, also termed Guanylyl Cyclase-A or GC-A), and CNP binds NPR type B (NPRB, also termed Guanylyl Cyclase-B or GC-B) [9–11]. The NP clearance receptor (NPRC) inhibits the effects of NPs by binding and degrading ANP, BNP and CNP [10–13]. The negative regulatory actions of NPRC play an important role at inhibiting the effects of NPs in podocytes because: 1. Podocytes express NPRA, NPRB and NPRC [12,14–17], and 2. NPRC is highly expressed in glomerular podocytes compared to other glomerular and kidney cell types [12,14–17].

Recent studies suggest that NPs have beneficial actions in glomerular diseases [15,17–22]. For example, KO of the cGMP generating ANP/BNP receptor (NPRA) exacerbates glomerular injury in proteinuric mouse models [15,22]. Moreover, TG overexpression of BNP ameliorates diabetic kidney disease [18] and immune medicated renal injury [21] in mouse models. These beneficial effects of NPs are mediated, at least in part, by stimulating cGMP generation [9,20]. This increase in cGMP generation inhibits signaling pathways that play important roles in kidney diseases including calcium signaling, Rho GTPases and TGF-beta [9,20].

We hypothesized that inhibition of NP clearance by NPRC would increase the local concentration of NPs, increase cGMP signaling in glomerular podocytes and inhibit glomerular injury. In support of this hypothesis, we previously found that: 1. NPs potently stimulated cGMP generation and inhibited podocyte apoptosis in cultured podocytes [17], and 2. Pharmacologic blockade of NRPC in vivo enhanced urinary cGMP excretion and significantly reduced albuminuria in a mouse model of FSGS [17]. To further test the hypothesis in vivo, we deleted NPRC [9] specifically in glomerular podocytes in a transgenic (TG) mouse model of FSGS created in our laboratory [23], This model sensitizes mice to the podocyte toxin puromycin aminonucleoside (PAN). In these TG mice, a single dose of PAN induces GS and heavy proteinuria. We found that podocyte specific NPRC KO significantly reduced albuminuria, GS and TI inflammation, inhibited both myofibroblast activation and deposition of fibronectin and collagen 4 (alpha-1 chain) in glomeruli, and preserved expression of the podocyte proteins nephrin and podocin. These data suggest that inhibition of NP clearance by NPRC may be a useful therapeutic approach to treat FSGS and perhaps other glomerular diseases.

## Results

### Podocyte protective effects of NPs in cultured podocytes

We previously found that both ANP and CNP protected cultured podocytes from apoptotic stimuli [17]. ANP is mainly produced by the heart and acts in an endocrine fashion to affect the function of target organs [24]. In contrast, CNP is produced and acts locally in an autocrin and paracrine fashion [24]. There is strong evidence that that ANP has potent podocyte protective actions in animal models of glomular disease [15,17–19,21,22], but little is known about the paracrine/endocrine actions of CNP in the kidney.

CNP is expressed in both the tubules and glomeruli by in-situ hybridization [25]. In glomeruli, CNP was detected in both the parietal and visceral layers of the glomerulus [25]. The presence of CNP mRNA in the viseral layer suggests that CNP is expressed in podocytes (visceral epithelial cells). Consistent with these findings, CNP was detecded in cultured mouse podocytes by RT-PCR [12]. Moreover, the CNP receptor (NPRB) is expressed by podocytes [12,14,16,17] and stimulates cGMP generation [12,17]. Taken together, these data indicate

that CNP production by glomerular podocytes might act in an autocrine, podocyte protective fashion as depicted in Figs 1A and 1B.

We first determined if inhibiting CNP clearance by knockdown (KD) of NPRC increased the autocine podocyte protective effects of CNP. For the studies, we used 4 shRNA constructs (1, 2, 3 & 4) and a scrambled control construct to KD NPRC. As shown in Fig 1C, constructs 1 & 2 provided the most effective KD of NPRC after selection, and these cell lines were chosen for the apoptosis studies. To further confirm KD, we examined the effect of low dose CNP on cGMP generation in cultured mouse podocytes. Given that podocytes express high levels of neprilysin [26,27], we measured CNP in the presence or absence of the neprilysin inhibitor LBQ657 [28]. As shown in Fig 1D and 1E, there was a significant increase in CNP levels in NPRC KD cells compared to wild type (WT) podocytes in both the presence and absence of LBQ657. We also examined the effect of NPRC KD on NPRB signaling by measuring cGMP generation (Fig 1F). In KD cells, cGMP generation was significantly increased compared to control cells. Lastly, we evaluated: 1. The effects of serum deprivation on apoptosis in NPRC-KD cells and controls, and 2. The effect of pharmacologic blockade of the CNP receptor (NPRB) on the apoptotic response. As shown in Fig 1G, serum deprivation significantly increased apoptosis in control cells, and the apoptotic response was significantly reduced in the KD podocytes in the absence of adding exogenous NPs to the culture medium. Moreover, pharmacologic blockade of NPRB with P19 [29,30] attenuated the podocyte protective effect

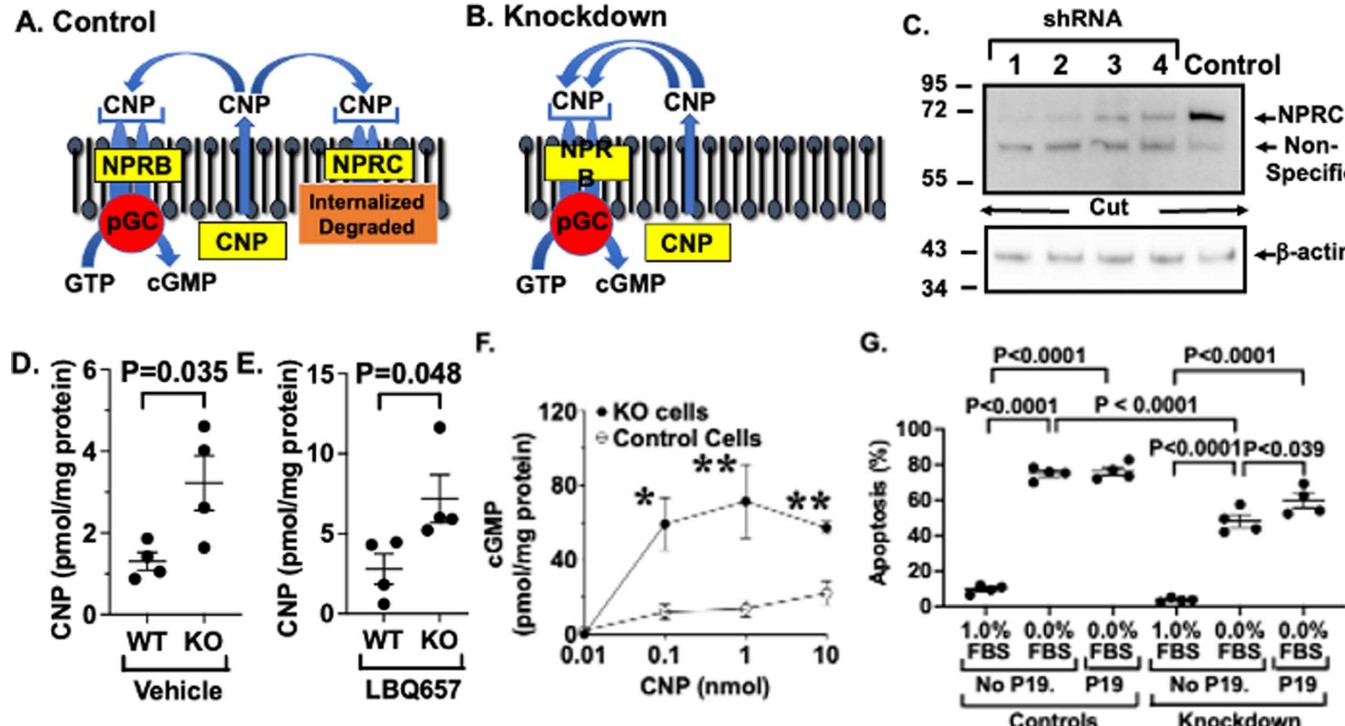

**Fig 1. (1A) CNP is produced by podocytes and binds to both NPRB and NPRC on glomerular podocytes.** (1B) Knockdown (KD) of NPRC increases CNP levels by inhibiting clearance of CNP from the circulation. (1C) Four shRNA constructs (1, 2, 3 & 4) and a scrambled control were used to KD NPRC in cultured podocytes. Constructs 1 and 2 provided the most effective KD of NPRC. (1D & 1E) CNP expression in cell culture medium was significantly increased by KD of NPRC in the both the presence and absence of the neprilysin inhibitor LBQ657. (1F) Generation of cGMP by low dosages of CNP was enhanced by NPRC KD in cultured podocytes compared to controls. (1G) Serum deprivation significantly increased apoptosis in control podocytes and KD of NPRC inhibited the apoptotic response in the absence of exogenous NPs. In addition, treatment with the NPRB antagonist P19 inhibited the podocyte protective actions of NPRC KD, resulting in an increase in apoptosis. *P < 0.005 **P < 0.001 versus control cells.

in NPRC KD cells. Lastly, similar beneficial effects of NPRC KD on apoptosis were observed using a shorter duration of serum deprivation (S1 Fig in S1 Data).

We next examined expression of NPRs in cultured podocytes and compared the podocyte protective effects of both ANP and CNP. Fig 2A shows NPRA, NPRB and NPRC mRNA expression in cultured podocytes. KD of NPRC effectively reduced NPRC mRNA in cultured podocytes (P < 0.0001). In addition, NPRC was highly expressed in WT podocytes compared to either NPRA or NPRB in the presence of absence of NPRC KD (P < 0.0001). Fig 2B demonstrates the differences in NPRA and NPRB expression by enlarging the vertical axis (ordinate) of Fig 2A. The mRNA levels of NPRA and NPRB were similar in cells expressing NPRC, and NPRA expression levels were not significantly changed by NPRC KD. NPRB was, however, significantly decreased in NPRC KD cells. Fig 2C shows the effects of 10 nM ANP and 10 nM CNP on cGMP generation. There was a stepwise increase in cGMP generation in both WT and KD podocytes. At the 10 nM concentrations, the increase in cGMP generation was statistically significant in the KD podocytes stimulated with either ANP or CNP compared to unstimulated KD podocytes and compared to WT podocytes expressing NPRC. In addition, we investigated the relative podocyte protective actions of ANP and CNP in cultured podocytes. As shown in Fig 2D, serum deprivation increased podocyte apoptosis and KD of NPRC reduced the apoptotic response in the absence of additional NPs added to the culture medium, similar to Fig 1G. Treatment with 10 nM ANP was, however, more effective at decreasing apoptosis in NPRC KD cells compared NPRC WT cells expressing NPRC. Moreover, combined ANP treatment with NPRC KD was the most effective approach to reduce apoptotic response. In contrast. treatment with 10 nM CNP alone had no significant effect on the apoptotic response in podocytes expressing NPRC, although NPRC KD inhibited podocyte apoptosis with or without the addition of 10 nM CNP. Taken together, these data suggest that

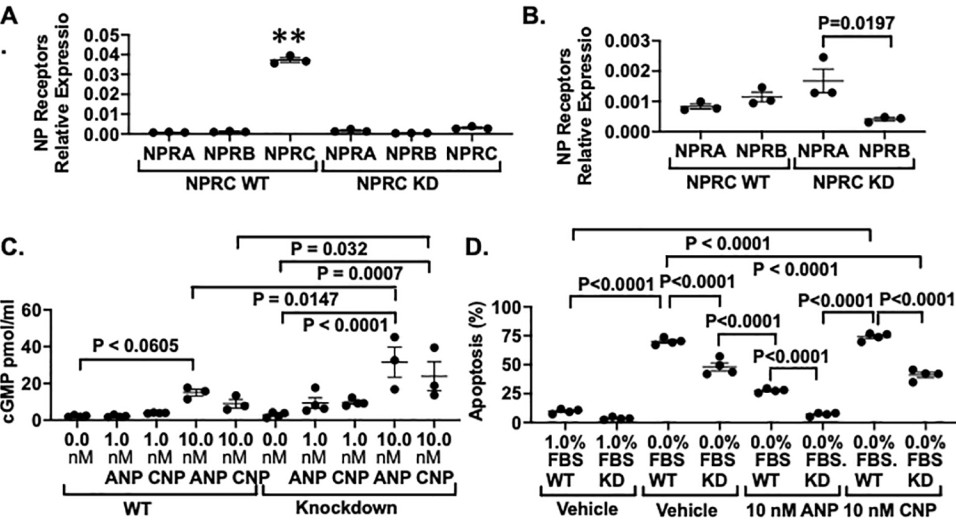

**Fig 2. (A) NPRC was highly expressed in WT podocytes compared to either NPRA or NPRB in the presence of absence of NPRC KD.** In addition, NPRC KD effectively decreased NPRC mRNA in cultured podocytes. (B) NPRB expression was downregulated in NPRC KD cells. (C) Both ANP and CNP increased cGMP generation in a stepwise fashion. The increase in cGMP generation was statistically significant for the KD cells at the 10 nM concentrations compared to both unstimulated KD cells and WT podocytes stimulated with either ANP or CNP. (D) Apoptosis induced by serum deprivation was inhibited by KD of NPRC. ANP treatment alone further inhibited podocyte apoptosis and combined treatment with ANP and NPRC KD reduced the apoptotic response toward baseline. In contrast, CNP alone had little effect on podocyte apoptosis, but NPRC KD reduced the apoptotic response in the presence or absence of 10 nM CNP. **P < 0.0001 versus other NPRs shown in Fig 2A.

ANP is more effective than CNP at inhibiting the apoptotic response and combining NPRC KD with ANP treatment further augmented the podocyte protective effect.

## Podocyte specific KO of NPRC in a mouse model of FSGS

To determine if inhibition of NP clearance had beneficial effects in a proteinuric kidney disease, we deleted NPRC specifically in glomerular podocytes in a TG mouse model of FSGS created in our laboratory [23]. As described in the Methods Section, these "double" TG mice express a doxycycline inducible, constitutively active Gq alpha-subunit (GqQ > L) specifically in podocytes [23]. Following induction of GqQ > L, a single dose of PAN causes robust albuminuria in Gq mice, but only mild disease in WT mice. To knockout NPRC in podocytes, Gq mice were crossed with TG mice expressing: 1. Two NPRC "floxed" alleles flanking exon 3 of NPRC [17], and 2. Cre-recombinase under the control of a doxycycline responsive promoter (see Methods). This created "triple" transgenic mice with 2 "floxed" NPRC alleles. In these animals, doxycycline induces podocyte specific GqQ > L expression and podocyte specific KO of NPRC on demand (Gq-KO mice). Control mice did not express GqQ > L and included WT NPRC KO mice (WT-KO mice) and WT mice expressing NPRC (WT mice).

We first determined the efficiency of podocyte specific NPRC-KO by crossing Gq mice with a reporter mouse expressing a two-color fluorescent Cre reporter allele [31]. The reporter allele contains a cell membrane-localized red fluorescence protein (tdTomato) with widespread expression in all tissues and cell types prior to induction of Cre recombinase. Expression of Cre recombinase induces a cell membrane-localized green fluorescence (EGFP) in cells expressing Cre recombinase, with red fluorescence confined to other cell types. As shown in Fig 3A, green fluorescence was induced in glomeruli of tdTomato mice following induction of Cre recombinase by doxycycline. Little EGFP expression was observed in mice expressing only the reporter construct (Fig 3B). A low power view of the tdTomato studies is shown Supplementary S2 Fig in S1 Data.

We also measured expression of NPRC mRNA and NPRC protein in enriched glomerular preparations from both WT mice and KO mice using quantitative RT-PCR and immunoblotting. As shown in Fig 3C, 3D and 3E, there was a significant decrease in both NPRC mRNA and NPRC protein in glomerular preparations from KO mice compared to WT mice. Expression of NPRC in kidney cortices was similar in WT and KO kidneys.

## Systemic BP and albuminuria

We next evaluated the effects of NPRC-KO on systemic blood pressure (BP) and proteinuria using the protocol described in the Methods Section (see schematic in Supplementary S3 Fig). As shown in Fig 4A, systolic BP was similarly reduced in both groups of Gq mice compared to controls. Although there is variability in the BP results, the BP is similar in the Gq mice (GQ and Gq-KO), suggesting that a difference in BP did not affect the severity of renal injury in the Gq groups. Fig 4B shows the effects of treatment with PAN on proteinuria in Gq mice. PAN induced heavy proteinuria on days 10 and 14 in Gq-mice compared to baseline. Podocyte specific KO of NPRC significantly reduced albuminuria at both the 10-day and 14-day time points. PAN had little effect on albuminuria in controls (supplemental S1 Table in S1 Data).

## Renal histopathology

For these studies, histopathologic abnormalities were graded by a pathologist blinded to genotype using previously described criteria [32] (see Methods). Figs 5A and 5B show representative pictures of GS in Gq mice and Gq-KO mice. As shown in Fig 5C, there was a significant decrease in GS in Gq-KO mice compared to Gq mice. A similar significant decrease was

observed for tubulointerstitial (TI) inflammation (Fig 5D). Lastly, tubular injury tended to be more severe in the WT Gq mice compared to Gq-KO mice (Fig 5E), but the overall difference was not statistically significant (Fig 5E). Representative low power pictures of the histopathology in the controls and Gq groups are shown in supplementary S4 & S5 Figs). In addition, the number of mice with mild, moderate and severe GS is reported in supplementary S6 Fig.

## Glomerular fibrosis

To further evaluate the effects of NPRC KO on glomerular fibrosis, we measured fibrotic markers in enriched glomerular preparations. For these studies, data from WT mice and WT-KO mice were similar and were combined for the analyses (WT controls). Figs 6A and 6B show expression of the myofibroblast differentiation marker alpha-smooth muscle actin (alpha-SMA) and the extracellular matrix protein (ECM) fibronectin. There was a statistically significant increase in glomerular expression of mRNA for alpha-SMA in Gq mice compared to WT controls, and KO of NPRC significantly reduced alpha-SMA in Gq-KO mice. A similar pattern was observed for fibronectin, but the decrease did not reach statistical significance. We also evaluated expression of the alpha-1 chains of collagen 4 (COL4 alpha-1) and collagen 1 (Col1 alpha-1) in the glomerular preparations (Figs 6C and 6D). Col4a1 is a major component of the FSGS lesion [33–35], and type 1 collagens are significantly increased in both global GS

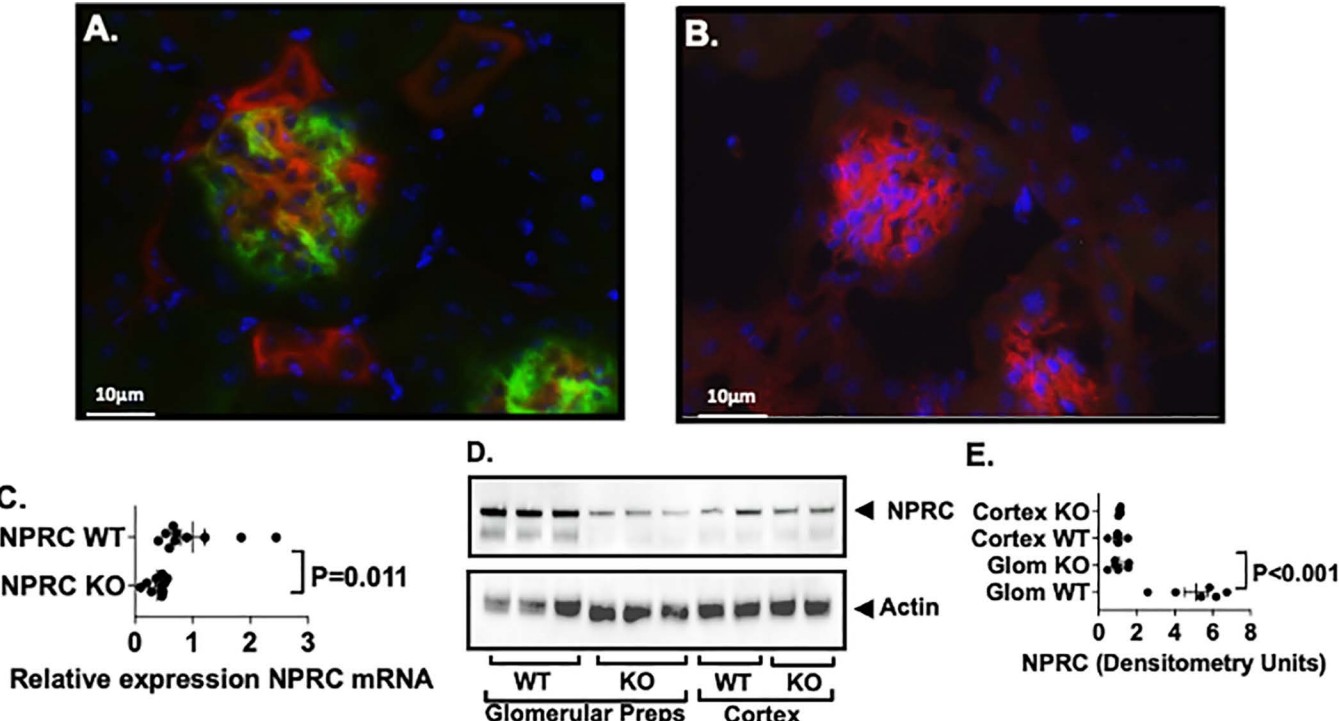

**Fig 3. Cre-mediated recombination in vivo. (3**A & **3**B). These studies used a mouse expressing a reporter allele that contains a cell membrane-localized red fluorescence transgene (td-Tomato) with widespread expression in all tissues and cell types. Expression of Cre recombinase induces a cell membrane-localized green fluorescence protein (EGFP). To examine Cre-medicated recombination in vivo, "double" TG mice (NPHS2-rtTA, tetO-Cre) mice were crossed with the TG reporter animals to create triple TG mice (NPHS2-rtTA, tetO-Cre, td-Tomato). In these TG mice, treatment with doxycycline induces green fluorescence specifically in podocytes, with red fluorescence confined to other cell types (panel **3**A). In contrast, little EGFP expression was observed in controls (panel **3**B). (**3**C) Quantitative RT-PCR demonstrated a significant decrease in expression NPRC in enriched glomerular preparations from NPRC KO mice compared to controls. (3D & 3E). NPRC protein was significantly reduced in glomerular preparations form podocyte specific KO mice compared to WT mice. Expression of NPRC in kidney cortices was similar in WT and KO kidneys.

[36,37] and TI fibrosis [34,38]. As shown in Fig 6C, there was a significant increase COL4 alpha-1 mRNA in glomerular preparations from Gq mice compared to WT controls, and this increase in COL4 alpha-1 in Gq mice was significantly inhibited in Gq KO mice. In contrast, there were no significant differences in expression of COL1 alpha-1 in the glomerular preparations (6D).

To confirm the mRNA studies, we evaluated expression of fibrotic markers including the myofibroblast marker alpha-SMA and the extracellular matrix protein fibronectin in glomerular preparations by immunoblotting. As shown in Figs 5E and 5F, expression alpha-SMA was significantly increased in glomerular preparations from Gq mice compared to WT controls, and this increase was significantly reduced by KO of NPRC. Similarly, fibronectin expression was significantly increased in glomerular preparations from Gq mice compared to WT controls and this increase in fibronectin deposition was significantly reduced by KO of NPRC (Figs 6G and 6H).

### Podocyte protective effects of NPRC KO

In addition to fibrotic markers, we also examined the effect of podocyte specific KO of NPRC on the podocyte proteins nephrin and podocin. As shown in Figs 7A and 7B, expression of both nephrin and podocin mRNAs were significantly decreased in glomerular preparations from Gq mice, and podocyte specific KO of NPRC significantly increased nephrin and podocin mRNA expression in Gq-KO mice compared to Gq mice. We next measured nephrin protein levels in glomerular preparations by immunoblotting. As shown in Figs 7C and 7D, the pattern of nephrin expression was similar to the quantitative RT-PCR results, but the changes were not statistically significant. Additional mRNA data is presented for WT1, synaptopodin, podocalyxin, TGF-beta and IL11 in Supplementary S2 Table in S1 Data.

Lastly, we evaluated kidney function in Gq mice and controls by measuring cystatin C levels in serum, which is reported to be more precise for assessing renal function in mice compared to serum creatinine levels [39]. As shown in Fig 8, there was a significant increase in cystatin C levels in Gq mice compared to controls. Podocyte specific KO of NPC reduced cystatin C levels compared to Gq mice, but the decrease was not statistically significant.

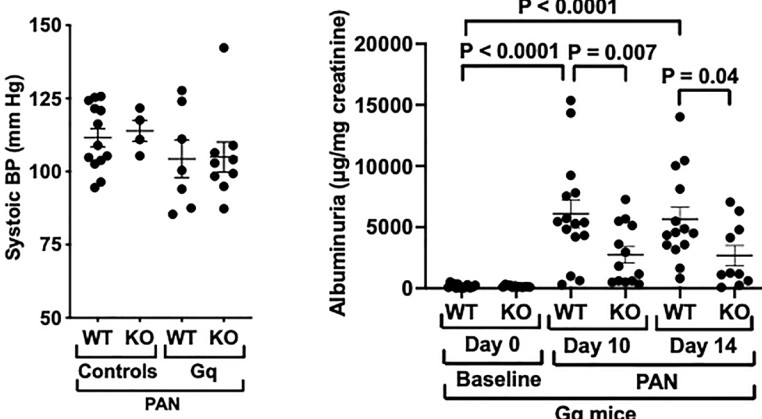

**Fig 4. Effect of podocyte specific NPRC KO on systemic BP and albuminuria.** (4A) Systolic BP was similarly decreased in both groups of Gq mice compared to controls, but this difference was not statistically significant. (4B) Treatment with PAN induced heavy proteinuria at day 10 and day 14 in WT Gq mice. The increase in albuminuria was significantly decreased by NPRC-KO at both day 10 and day 14 following PAN injection.

## Discussion

In this study, we found that podocyte specific KO of NPRC had podocyte protective effects in cultured podocytes and ameliorated kidney disease in a mouse model of FSGS. In the kidney, KO of NPRC: 1. Decreasing albuminuria, GS, and TI inflammation, 2. Preserving expression of the podocyte proteins nephrin and podocin, and 3. Inhibiting expression of the myofibroblast marker alpha-SMA and the ECM proteins fibronectin and Col4 alpha-1. The beneficial effects of NPRC KO were accomplished without a significant difference in systemic blood pressure between Gq mice and Gq-KO mice. These data suggest that podocyte specific KO of NPRC had multiple beneficial effects in an FSGS model. Taken together, we posit that

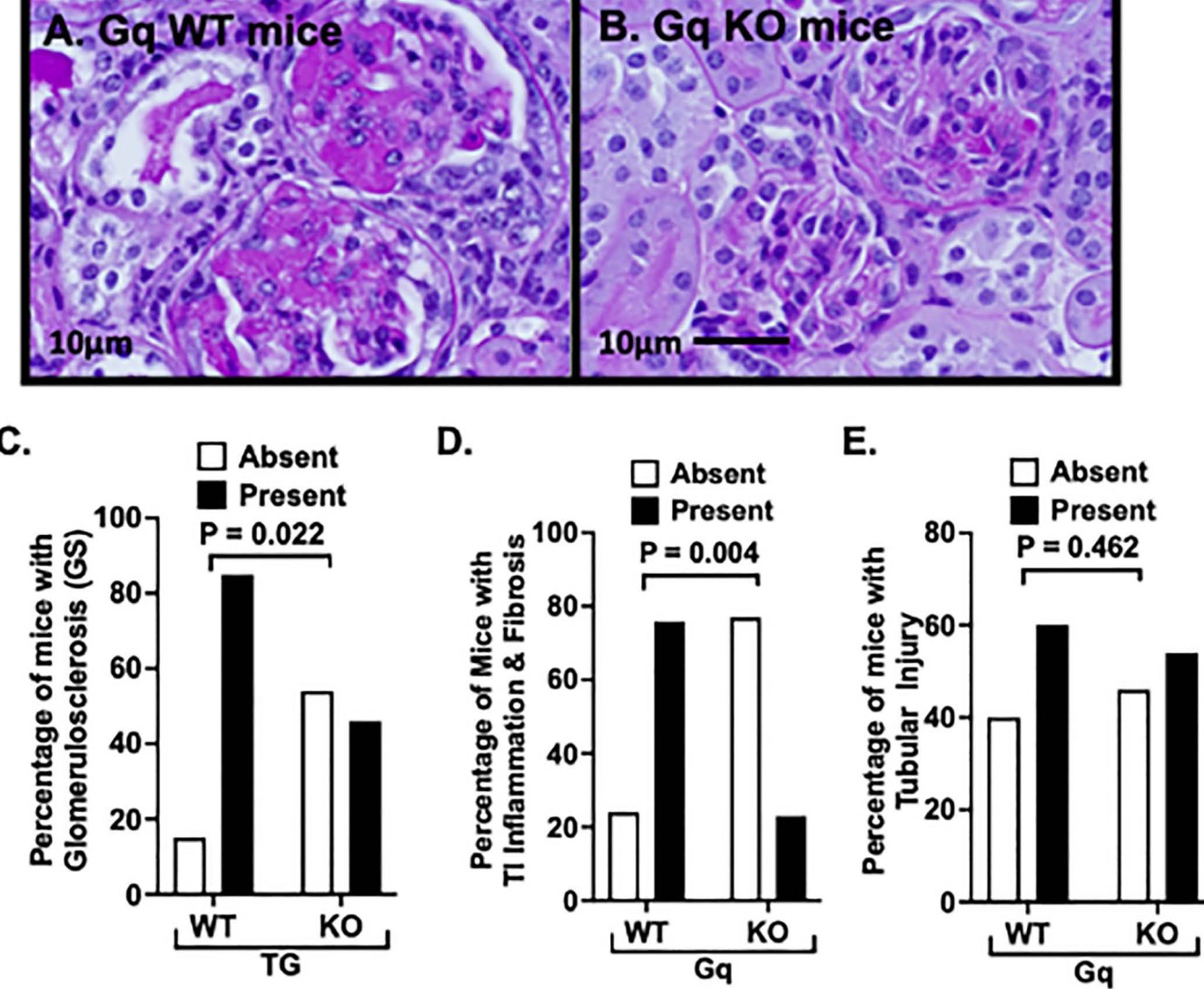

**Fig 5. Effect of podocyte specific NPRC KO on renal histopathology.** (5A, 5B) Representative pictures of GS in Gq mice and Gq-KO mice. (5C) NPRC-KO significant reduced GS in Gq-KO mice compared to Gq mice. (5D).NPRC-KO significantly reduced TI inflammation in Gq-KO mice compared to Gq mice. (5E) Tubular injury was similar in both groups of Gq mice. (NOTE: data in Figs 5C, 5D and 5E are presented as a percentage of mice with the indicated histologic abnormalities, but the statistical evaluation was performed using the number of mice with the specified histologic abnormality).

inhibiting NP clearance to enhance the effects of endogenous NPs may be a useful therapeutic approach to treat glomerular disease processes.

The biological effects of NPs are modulated by multiple mechanisms in addition to clearance of NPs by NPRC including: 1. The protease neprilysin that cleaves circulating NPs [11], and 2. Intracellular hydrolysis of cGMP by phosphodiesterases (PDEs) [11,40]. All these mechanisms likely play significant roles in regulating the biologic effects of NPs. We previously found that: 1. NPRC and neprilysin are expressed in cultured podocytes [17], in agreement with previous studies [8,14,15,17,27], 2. Cultured podocytes express PDEs important for regulating intracellular levels of cGMP including PDE5 and PDE9 [17], and 3. Pharmacologic blockade of NPRC in cultured podocytes enhanced cGMP generation to a greater extent than pharmacologic inhibition of either PDE5, PDE9 or neprilysin inhibition, using maximally effective doses of each drug [17]. These results agree with in vivo studies that demonstrated pharmacologic blockade of NPRC increased ANP more effectively than neprilysin antagonists [10,17,41–44]. These data suggest that targeting NPRC is an effective strategy to reduce clearance of NPs from the circulation by NPRC and enhance NP signaling.

Both ANP and BNP bind to NPRA, stimulate cGMP generation, and have renal protective actions in animal models of kidney disease [15,18,21,22]. Less is known about potential renal protective effects of CNP. Moreover, CNP is expressed by multiple cell types including podocytes (visceral epithelial cells) [12,25] and may have local renal protective actions. In cultured podocytes, we found that KD of NPRC enhanced cGMP generation by low doses of CNP (Fig 1F) and reduced apoptosis induced in the absence of treatment with exogenous NPs (Fig

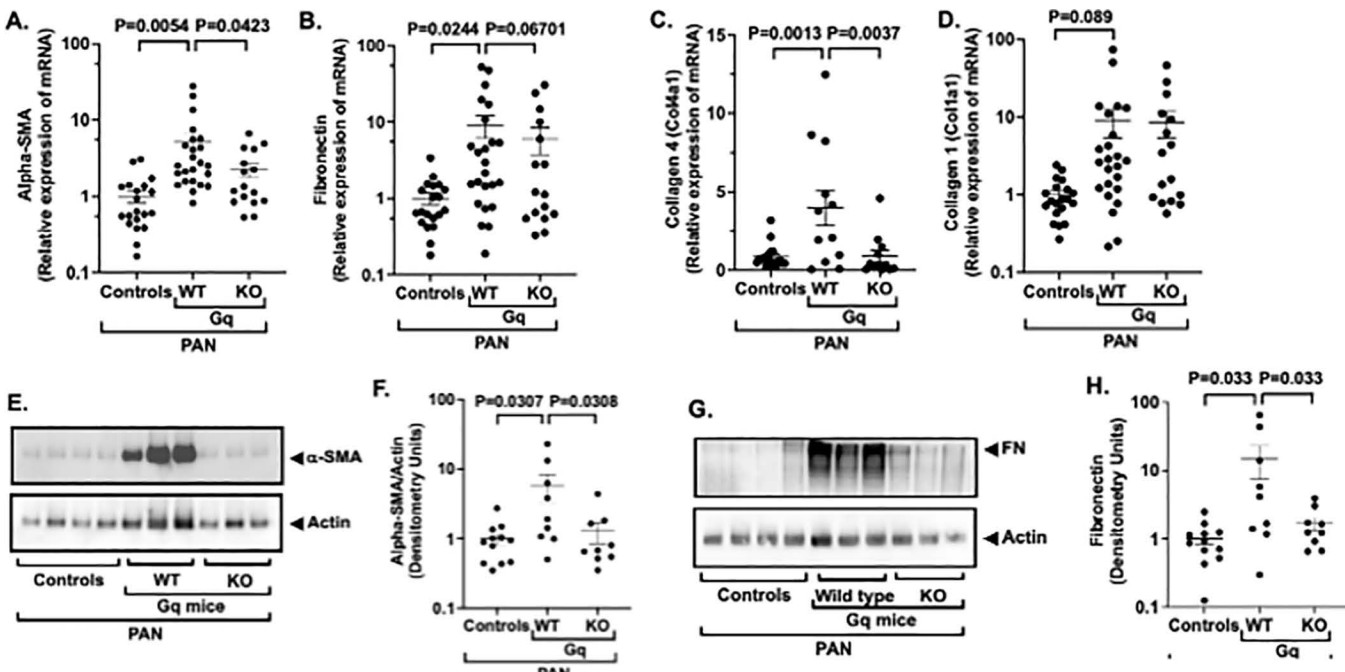

**Fig 6. Effect of NPRC KO on fibrotic markers.** (6A) The myofibroblast marker alpha-SMA was significantly increased in glomerular preparations from WT Gq mice compared to controls. The increase in alpha-SMA was significantly reduced by podocyte specific KO of NPRC. (6B, 6C & 6D) The ECM proteins fibronectin and the alpha-1 chains of collagen 4 (CO4 alpha-1) and collagen 1 (Col1 alpha-1) were significantly increased in Gq mice compared to controls. Podocyte specific KO of NPRC decreased expression of both fibronectin and Col4 alpha-1, which was statistically significant for Col4 alpha-1. In contrast, NPRC KO had little effect on Col1 alpha-1 expression. (6E, 6F, 6G & 6H) Protein levels of the myofibroblast marker alpha-SMA and fibronectin were significantly increased in glomerular preparations from WT Gq mice compared to controls. Podocyte specific KO of NPRC significantly reduced expression of both alpha-SMA and the ECM protein fibronectin. NOTE: Quantitative RT-PCR data is presented in logarithmic format.

1G). The podocyte protective action of NPRC KD was partially attenuated by pharmacologic blockade of NPRB in KD cells using the NPRB antagonist P19 (Fig 1G) [29,30]. Interpretation of the P19 experiments is, however, dependent on the specificity of P19 blockade [29]. In the KD cells, binding of P19 to either NPRC or NPRA is unlikely to affect the apoptotic response because: 1. NPRC is absent in the KD cells, and 2. The dosage of P19 used in the studies (250 nM) effectively binds NPRB (Kd 15.4 nM) but has significantly less affinity for NPRA (NPRA Kd 575 nM). In contrast to the KD cells, NPRC is expressed in the control cells, which could affect the apoptotic response by blocking natriuretic clearance and enhancing CNP levels. We posit that blockade of NPRC in the control cells (Fig 1G) was little affected by P19 treatment because P19 has high affinity for both NPRB (Kd ~ 15 nM) and NPRC (0.0134 nM) and, in turn, would effectively inhibit both NP receptors (NPRB and NPRC). In this scenario, the benefits of increasing CNP levels by blockade NPRC would be reduced by pharmacologic blockade of NPRB. Taken together, these data suggest that the podocyte protective actions of NPRC KD are mediated, as least in part, by endogenously produced CNP acting in an autocrine fashion to stimulate NPRB signaling.

We also acknowledge that that NPRC is not only a clearance receptor, but also stimulates intracellular signaling. Following ligand engagement, NPRC stimulates phospholipase C and inhibits both L-type ion channels and cAMP generation [45–47]. Theses signaling pathways have been reported to play important roles in cardiovascular disease [46,48], although their

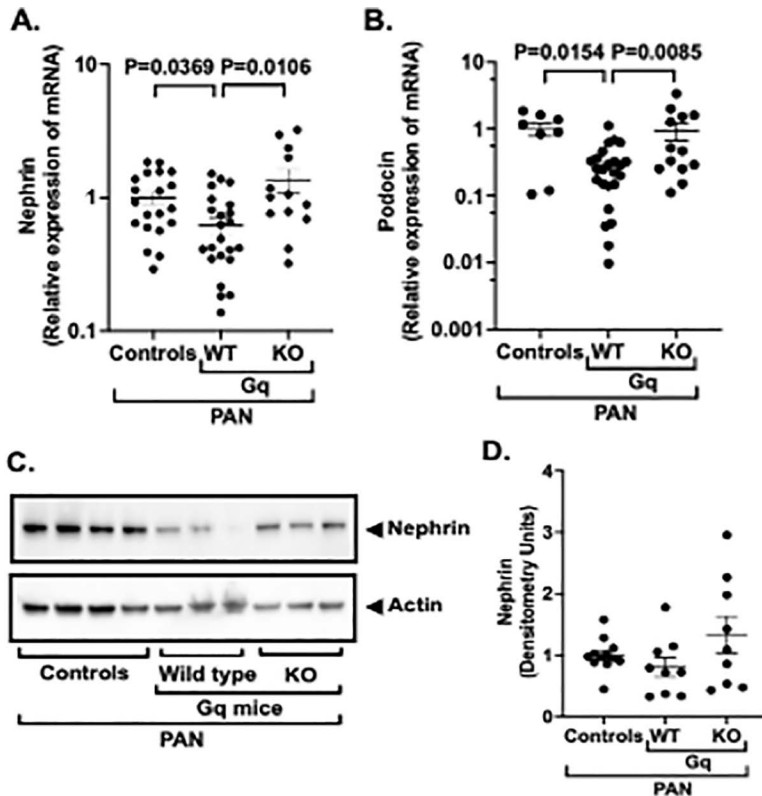

**Fig 7. Effect of NPRC KO on podocyte proteins.** (7A & 7B) Expression of nephrin and podocin mRNAs were significantly decreased in glomerular preparations from Gq mice compared to controls and expression of both podocyte proteins were preserved by podocyte specific KO of NPRC. (7C & 7D) The pattern of nephrin protein expression was similar to the mRNA results, but the differences were not statistically significant.

role in kidney diseases has not been carefully investigated. As a result, signaling pathways linked to NPRC activation may also contribute to the beneficial effects of NPRC KO.

In addition, we compared the effects of ANP and CNP on both cGMP generation and apoptotic responses. We found that activation of NPRA by ANP was more effective than CNP at inhibiting podocyte apoptosis at equivalent 10 nM concentrations in cultured podocytes (Fig 2D). While we can only speculate about mechanisms, previous studies suggest that ANP and CNP receptors bind their respective ligands with similar, high affinities [11,49] and, while single cell sequencing studies suggest that NPRA expression is more abundant in podocytes compared to NPRB [14,16,50], the levels of NPRA and NPRB were similar in cultured podocytes (Fig 2B) with the exception of the decrease in NPRB levels in the NPRC KD cells (Fig 2B) (discussed below). The culture system does, however, have significant differences from the in vivo situation. CNP is produced by podocytes [12,25] and CNP levels in culture medium were enhanced in NPRC KD cells (Figs 1D, 1C). Chronic exposure to NPs causes homologous desensitization [11,51], which causes a decrease in NPR responsiveness. In this scenario, continuous stimulation of NPRB by CNP may attenuate NPRB signaling. Moreover, prolonged exposure to ligand could theoretically cause receptor downregulation [51]. While downregulation of NPRs is controversial [10,52], recent studies suggest that prolonged treatment with ANP downregulates NPRA [53]. In a similar fashion, chronically enhanced CNP levels might explain the decrease in NPRB levels (Fig 2B) in cultured NPRC KD podocytes (Fig 2B).

To determine if enhancing NP signaling was podocyte protective in vivo, we examined the beneficial effects of NPRC KO in a mouse model of FSGS. We found that podocyte specific KO of NPRC reduced proteinuria, improved renal histology and decreased expression of the myofibroblast marker alpha-SMA and the ECM proteins fibronectin and Col4 alpha-1. The signaling pathways that regulate expression of these fibrotic markers are complex, but NPs inhibit several pathways that promote fibrosis. For example, NPs are potent inhibitors of Rho

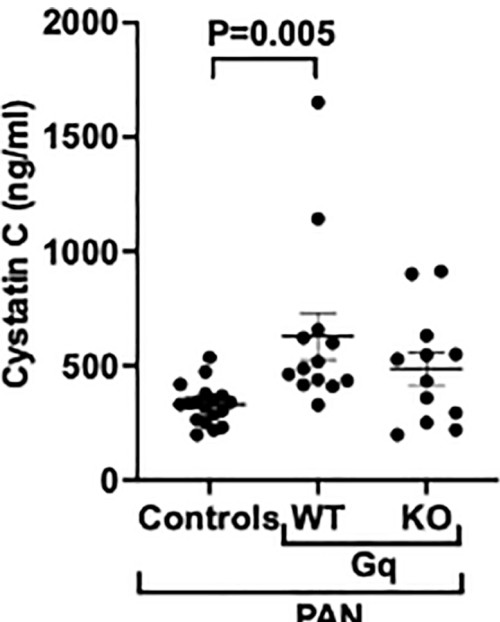

**Fig 8. Effect of NPRC KO on renal function.** Cystatin C levels were significantly increased in serum of Gq mice compared to controls. KO of NPRC decreased cystatin C levels in Gq mice but the difference was not statistically significant.

A [54], which decreases both fibroblast proliferation [11] and differentiation of fibroblasts into myofibroblasts [54]. In addition, cGMP signaling has diverse effects on calcium signaling [9] including: 1. Directly phosphorylating and inhibiting ion channels that increase intracellular calcium levels such as TRPC6 [55–58], 2. Sequestering calcium in the sarcoplasmic reticulum [59], and 3. Inhibiting calcium release from intracellular stores [60,61]. As a result, calcium signaling is impaired in multiple receptors implicated in promoting renal fibrosis including angiotensin II and endothelin [9,11,54,62,63]. Lastly, NPs also inhibit the master regulator of fibrosis, TGF-beta, by phosphorylating SMADs on an inhibitory site [20]. Thus, multiple potential mechanisms may contribute to the beneficial effects of NP signaling on renal fibrosis.

As mentioned above, NPRC KO decreased the expression of the ECM protein fibronectin (Figs 6G & 6H), but had only a modest, and non-significant effect on fibronectin mRNA expression (Fig 6B). While fibronectin is produced within the glomerulus, the protein also circulates in blood and is deposited in glomeruli in kidney diseases including FSGS [64] and diabetic kidney disease [65]. The predominant source of circulating fibronectin is the liver [64] and, in animal models, the major source of fibronectin in early stages of FSGS is from the circulation with deposition of glomerular fibronectin occurring later in the disease process [64]. Given the duration of the current animal studies (14 days), the increase in glomerular fibronectin in the present study was likely caused by deposition of fibronectin from the circulation, with lesser amounts produced in the glomeruli. We suspect that the discrepancy in the mRNA expression and protein levels is related to greater deposition of fibronectin from the circulation in damaged glomeruli compared to glomeruli with less severe glomerular injury.

We also found a significant decrease in TI inflammation in Gq-KO mice (Fig 5D). A similar improvement in the accumulation of inflammatory cells was observed by stimulating soluble guanylate cyclase and, in turn, enhancing cGMP generation in a rat model of anti-Thy1-induced glomerulonephritis [66]. Indeed, the same signaling pathways that stimulate myofibroblast differentiation (TGF-beta, Rho GTPases, and calcium) may also promote inflammation by modulating immune cell migration and the severity of immune response [67–69]. The level of proteinuria may also contribute to TI inflammation. In this regard, filtered proteins are resorbed by the proximal tubules and processed by lysosomes and endoplasmic reticulum [70]. In the setting of heavy proteinuria, these intracellular pathways are stressed [71], which results in the secretion of cytokines that both attract and activate inflammatory cells [72] and produce profibrotic mediators such as TGF-beta [73]. These data suggest that the anti-inflammatory actions of the NPRC inhibition may play an important role in the beneficial actions of this treatment approach.

The evidence that NPRC clears NPs from the circulation was initially based on: 1.Pharmacologic blockade of NPRC increased ANP plasma levels in rats [74], and 3. KO of NPRC increased the half-life of ANP in the circulation in mice [13]. These data suggested that a major function of NPRC was to negatively regulate the effects of ANP. More recent studies indicate that other NP family members are negatively regulated by the clearance receptor (NPRC). For example, mutations that disrupt the function of genes encoding either CNP or NPRB (CNP receptor) causes dwarfism [75–77]. In contrast, disruption of the gene encoding the clearance receptor (NPRC) caused bone overgrowth [13]. These findings indicate: 1. A major function of CNP is binding to NPRB and stimulating bone growth, and 2. The stimulatory effects of CNP on bone growth are augmented by decreased clearance of CNP from the circulation by NPRC. Taken together, these data are consistent with the notion that multiple NP family members are negatively regulated by NPRC.

Lastly. there have been several clinical studies examining the effects of enhancing NP signaling on progression of chronic kidney disease (CKD) using a drug combination that

contained both the neprilysin inhibitor sacubitril and the angiotensin receptor blocker (ARB) valsartan [78–80]. Two of these manuscripts [78,79] were secondary analyses of studies in heart failure patients that compared sacubitril/valsartan to renin-angiotensin system inhibition alone. Both these secondary analyses found a significant reduction in the rate of decline in the glomerular filtration rate (GFR) in patients receiving the drug combination (sacubitril/valsartan) compared to the control group [78,79]. Interpretation of these studies is, however, complicated by the significant beneficial effects of sacubitril/valsartan on heart failure, which may have improved renal hemodynamics. A third study investigated the effects of sacubitril/valsartan versus the ARB irbesartan in patients with CKD using a randomized, double-blind study design [80]. In this trial, treatment with sacubitril/valsartan caused a significant decrease in systolic and diastolic blood pressure but no clear improvement in GFR or proteinuria [80,81]. Interpretation of these results is complicated because of the short duration of the study, the low number of patients enrolled in the trial and the diverse study population that included a large number of patients without glomerular diseases.

In summary, knockdown of NPRC enhanced the podocyte protective actions of both ANP and CNP in cultured podocytes, and podocyte specific KO of NPRC had beneficial effects on proteinuria, GS, TI inflammation and expression of slit diaphragm and ECM proteins in a mouse model of FSGS. These data suggest that inhibiting NP clearance by NPRC might be a useful therapeutic approach to treat glomerular diseases.

## Methods

Primary antibodies used for the studies included: 1. A mouse monoclonal antibody to alpha-smooth muscle actin [82] (clone 1A4, catalog number: A5228, Sigma-Aldrich, St. Louis, MO), 2. A mouse monoclonal antibody to actin [83] (clone C4, catalog number: MA1501, Sigma-Aldrich, St. Louis, MO), 3. A rabbit polyclonal antibody to fibronectin (catalog number: ab2413, Abcam biotechnology, Cambridge, United Kingdom), 4. A goat polyclonal antibody to nephrin [84] (catalog number: AF3159, R&D Systems, Minneapolis, MN), 5. A rabbit monoclonal antibody to WT1 (Wilms tumor 1) (catalog number: ab89901, Abcam Biotechnology, Cambridge, United Kingdom), 6. A mouse monoclonal IgG antibody to the podocyte marker synaptopodin (catalog number 65194, Progen Biotechnik, Heidelberg, Germany), 7. A mouse monoclonal IgG antibody to NPRC (clone OTI4H1, catalog number: TA501044, Origene Technologies, Rockville, MD) and a polyclonal Goat IgG antibody to CNP (AF3127, R&D Systems, Minneapolis, MN). Secondary antibodies used for the studies included: 1. A mouse HRP-linked anti-goat polyclonal antibody (catalog number: sc-2354, Santa Cruz Biotechnology, Dallas, TX), and 2. An anti-mouse HRP-linked polyclonal antibody (catalog #: 7076) and an anti-rabbit HRP-linked polyclonal antibody (catalog #: 7074) from Cell Signaling Technology, Danvers, MA). 3. A Cy3 labeled donkey polyclonal anti-goat IgG antibody (Abcam Biotechnology, Cambridge, United Kingdom), and 4. A FITC labeled polyclonal donkey anti-mouse IgG antibody (Abcam Biotechnology, Cambridge, United Kingdom). Additional materials included: 1. Albuwell and Creatinine Companion kits (catalog No. 1011 and 1012, respectively, Ethos Biosciences, Logan Township, NJ), 2. Rat CNP (Item No. 24276) and cGMP ELISA kits (Item No.581021) from Cayman Chemical, Ann Arbor, MI, 3. Annexin V apoptosis kits, (catalog No. 559763, BD Pharmingen, Franklin Lakes, NJ), and 4. Recombinant mouse IFN-gamma (catalog No. 485-MI-100, R&D Systems, Minneapolis, MN).

### BP measurements

Systolic BP was measured using a computerized tail-cuff system (Hatteras Instruments, Cary, NC, USA) in conscious mice as previously described [85]. To reduce variability in the results,

mice were acclimated to the experimental conditions for a week prior to the BP measurements. This technique has previously been shown to correlate closely with intra-arterial measurements [86].

## Animal subjects

The following mouse models were used for the studies: 1. GqQ > L mice were created in our laboratory as described [23], 2. (NPRC-flox/flox mice were obtained from Taconic Biosciences, No. 13105) and bred onto the FVB background for over 6 generations, 3. A doxycycline inducible Cre recombinase mouse line was obtained from Dr. Jeffery Kopp on the FVB background [87] (NOTE: This mouse are now available from Jackson Labs (tetO-Cre, No. 008205). The tdTomato mouse [31] was originally obtained from Jackson Labs (Stock No. 007576).

## Animal experiments

Animal experiments were performed using a transgenic (TG) mouse model of FSGS developed in our laboratory (Gq mice) [23]. The model was created by generating mice with 2 transgenes: 1. The reverse tetracycline transactivator (rtTA) under the control of the podocyte specific podocin promoter (NPHS2-rtTA, Jackson Labs, catalog No. 008202), and 2. GqQ > L under the control of the doxycycline inducible tet operator sequence (tetO) and a minimal cytomegalovirus (CMV) promoter (tetO-GqQ > L). To create a podocyte specific NPRC KO in this TG model, we bred GqQ > L mice with: 1. Animals expressing two NPRC floxed alleles (NPRC-f/f mice, Taconic Biosciences, No. 13105), and 2. A doxycycline inducible Cre recombinase line from Jackson Labs (tetO-Cre, No. 008205). This breeding strategy generated TG mice on the FVB/NJ background expressing 3 transgenes (NPHS2-rtTA, tetO-Cre, tetO-GqQ > L). In "triple" TG mice, treatment doxycycline in 2% sucrose water induces podocyte specific GqQ > L expression and NPRC-KO on demand. All animals were breed onto the FVB/NC background for over 6 generations. A schematic of the breeding strategy is shown in S7 Fig.

For the experiments, age and sex-matched male and female mice were used for the studies because the phenotype is similar in both sexes [17,23]. Mice were studied at 3-4 months of age and all mice were born within a 5-week period. Baseline twenty-four-hour urine collections were collected using metabolic cages specifically designed for collection of mouse urine (Hatteras Instruments, Cary, NC). After collecting baseline urines, mice were treated with doxycycline for 1 week and then nephrosis was induced by a single injection of PAN (500 mg/kg) by intraperitoneal (IP) injection. Doxycycline was continued for an additional 2 weeks. Systolic blood pressure (SBP) was measured as described [85] during the second week after the PAN injection, and following acclimation to the experimental procedure during the prior week. Repeat 24-hour urine collections were obtained at day 10 and day 14 following the PAN injection. Mice were euthanized after the last urine collection by injecting 250 mg/kg pentobarbital, IP, followed by bilateral thoracotomy. Blood was immediately obtained by cardiac puncture after euthanasia and then kidneys were removed, weighed and kidney tissue was saved for additional studies as described below. The experiments conformed to the *Guide for the Care and Use of Laboratory Animals* and were approved by the Duke and Durham VA Medical Centers' Institutional Animal Care and Use Committees. A schematic of the experimental protocol is provided in supplementary S3 Fig.

## Podocyte specific NPRC KO in td-tomato mice

To assess NPRC KO, we used a reporter mouse available from Jackson labs (Stock Nos. 007576) [85]. The reporter mice express a two-color fluorescent Cre reporter allele containing

a cell membrane-localized red fluorescence protein (tdTomato) with widespread expression in all tissues and cell types prior to Cre-mediated recombination (JAX: Stock Nos. 007576). Induction of Cre recombinase in podocytes induces a cell membrane-localized green fluorescence protein (EGFP). For the studies, we crossed the reporter mice with mice expressing NPHS2-rtTA and tetO-Cre to create mice expressing 3 transgenes (tdTomato, NPHS2-rtTA, and tetO-Cre). We then assessed the effectiveness of podocyte specific NPRC-KO by studying mice treated with either doxycycline or vehicle as described above.

## Histopathology

After fixation in formalin, light microscopic sections were stained with hematoxylin and eosin (H&E) and Masson trichrome by the Duke Research Animal Pathology Service. The slides were evaluated by an experience medical nephropathologist (A.F.B.) blinded to genotype. Each tissue section had more than twenty glomeruli for evaluation. Tubules were examined for tubule dilation and casts, and tubulointerstitial areas were examined for inflammation with or without interstitial fibrosis. There is little tubulointerstitial fibrosis observed in this model over a 14-day period, and this score was based on the severity of inflammation as described below. Abnormalities were graded using a semi-quantitative scale of 1-4 (1-normal, 2-mild, 3-moderate, 4-severe) as previously described [23,32] based on the following criteria:

Glomerulosclerosis.

1. Normal (baseline): None

2. Mild: < 10% of glomeruli

3. Moderate: 10-25% of glomeruli

4. Severe: > 25% of glomeruli

Tubule injury.

1. Normal (baseline): None to minimal tubular dilation and casts without tubular degeneration or regeneration

2. Mild: Tubular degeneration and regeneration with/without tubular dilation and casts, involving < 10% of cortex

3. Moderate: Tubular degeneration and regeneration with/without tubular dilation and casts, involving 10–25% of cortex

4. Severe: Tubular degeneration and regeneration with/without tubular dilation and casts, involving > 25% of cortex

Tubulointerstitial inflammation.

1. Normal (baseline): One focus of inflammation (up to 15 mononuclear cells) with or without interstitial fibrosis

2. Mild: Two foci of 15 + mononuclear cells or 1 focus of 30 + mononuclear cells involving up to 5% of area of cortical parenchyma with or without interstitial fibrosis

3. Moderate: Inflammation involving > 5–25% of the cortical parenchyma with or without interstitial fibrosis

4. Severe: Inflammation involving > 25% of the cortical parenchyma with or without interstitial fibrosis

## Immunofluorescence studies

Expression of human CNP in kidney sections was detected by immunofluorescence imaging using a goat polyclonal IgG antibody to human CNP (AF3127, R&D Systems), and a mouse monoclonal IgG antibody to the podocyte marker synaptopodin (Progen Biotechnik). Normal human kidney frozen sections were obtained from Zyagen (San Diego, CA) and were fixed in 2% paraformaldehyde for 5 minutes, air-dried, treated with 0.1% Triton-X in Dulbecco's phosphate buffered saline (D-PBS) for 5 minutes and then blocked for 1 hour in D-PBS with 5% bovine serum albumin (BSA). The CNP and synaptopodin primary antibodies were then added at a 1:200 dilution and 1:50, respectively in D-PBS with 5% BSA. After an overnight incubation, slides were washed 3 times in D-PBS and then incubated for 1 hour with a Cy3 labeled donkey polyclonal anti-goat antibody (Abcam Biotechnology), and a FITC labeled polyclonal donkey anti-mouse antibody (Abcam Biotechnology), both at a dilution of 1:1000 in D-PBS with 5% BSA. Sides were then washed 3 times in D-PBS, and coverslips were affixed using with Vectashield (Vector Labs, Newark, CA) mounting medium with DAPI (4',6-diamidino-2-phenylindole). Slide were examined by dual fluorescence microscopy using a Nikon Eclipse TB-2000 microscope.

## Cell culture studies

The immortalized mouse podocyte cell line was a gift of Dr. Paul E. Klotman, MD [88]. The cells were derived from animals bred with the H-2K$^b$-tsA58 Immortomice (Charles River Laboratories, Wilmington, MA). Podocytes were selected for expression of the podocyte markers WT-1, synaptopodin and podocalyxin. To permit immortalized growth, cells were grown at 33°C in medium supplemented with 10 units/ml gamma-interferon to induce the H-2K$^b$ promoter driving synthesis of the temperature sensitive (tsA58) SV-40 T antigen (permissive conditions). For differentiation, cells were grown at 37°C in medium lacking gamma-interferon, resulting in degradation of the T antigen (non-permissive conditions). Our lab further characterized differentiated cells by RT-PCR, and they also express low levels of the podocyte proteins nephrin and podocin. For the experiments, the immortalized mouse podocyte cell line was maintained in culture and was plated in either 6- or 12- well tissue culture clusters (Corning-Costar, Corning, NY) and differentiated for ~7 days prior to study as previously described.

To knockdown (KD) NPRC, we used four shRNA constructs (catalog numbers TG514036A-D; Origene Technologies, Rockville, MD) and a scrambled control (TR30014). The constructs were introduced into our mouse podocyte cell line using Lipofectamine 2000 according to the manufacturer's recommendations (catalog number 11668027, ThermoFisher Scientific, Waltham, MA), and then selected in 4µg/ml puromycin (catalog number: sc374043, Santa Cruz Biotechnology, Dallas, TX),

## Measurement of CNP in cell culture medium

For the study, KD cells and controls were plated in 12-well tissue culture clusters and differentiated for 7 days. After differentiation, cells were washed and placed in serum free medium warmed to 37°C, in the presence or absence of the neprilysin inhibitor LBQ657 (10 µM final) [28]. After incubation at 37°C for 30 minutes, the medium was aspirated, aliquoted into microfuge tubes, placed on ice and immediately stored at -80°C. CNP in culture medium was measured using an ELISA kit (RK02693) from ABcclonal technologies (Woburn, MA) and data was expressed as pmol/mg protein.

## Measurement cGMP in cell culture medium

For the cGMP studies, differentiated podocytes were made quiescent by incubation in serum free medium overnight. Cells were then treated with vehicle or the following concentrations of CNP (0.1, 1.0 and 10 nM). After 30 minutes, the supernatant was aspirated, and the tissue culture clusters placed on ice. Hydrochloric acid (0.1 N) was then added to the wells (100 μl per 12 wells and 300 μl per 6 well tissue culture cluster). A cell scrapper was used to remove the cells from the cell culture wells, and the mixture dissociated by pipetting up and down until homogeneous. The mixture was then centrifuged at 1000 x g for 10 minutes and the supernatant frozen at -80°C. Measurement of cGMP in the supernatants was performed using an ELISA kit from Cayman Chemicals (Ann Arbor, MI) according to the directions of the manufacturer and data was expressed as pmol/mg protein.

## Apoptosis studies

For the apoptosis studies, differentiated mouse podocytes were changed to medium with 1% fetal bovine serum (FBS) or serum free medium. After 2 days of serum deprivation, cells were harvested, and apoptotic cells were identified by annexin V staining using kits from BD Pharmingen (San Diego, CA) according to the directions of the manufacturer. Quantitation of the apoptotic cells was performed by flow cytometric analysis at the Duke Comprehensive Cancer facility. For the annexin V studies, apoptotic podocytes were differentiated from necrotic cells by staining with 7-amino-actinomycin D. For evaluation of these data, basal levels of apoptosis (cells treated with 0.1% FBS and vehicle) were subtracted from the results, and data expressed as the percent apoptosis above basal.

For the pharmacologic studies, P19 (250 nM) or vehicle was added to the culture medium. In addition, the culture medium with P19 or vehicle was replenished twice per day to minimize degradation of the P19 during the study.

## Immunoblotting of enriched glomerular preparations

Immunoblotting was performed as previously described [85] using the antibodies listed in Materials. Briefly, enriched glomerular pellets were solubilized by sonication in NP-40 lysis buffer (50 mM Tris-HCl, 150 mM sodium chloride, 2 mM ethylenediaminetetraacetic acid, 1% NP-40 and protease inhibitors from Sigma-Aldrich), and frozen at -80°C until the time of study. Proteins were separated using the XCell SureLock Bis-Tris Mini-Cell Electrophoresis System (Thermo Fisher Scientific, Waltham, MA) and transferred to PVDF (polyvinylidene fluoride) membranes according to the directions of the manufacturer. Immunoblots were then blocked in 20 mM Tris-HCl, 137 mM NaCl, pH 7.6 (TBS) with 0.2% Tween 20 (T-TBS) and 2% bovine serum albumin. The primary antibody was added at a 1:1000 dilution in blocking buffer and incubated overnight. After washing, the HRP (horseradish peroxidase) linked secondary antibodies were added at a 1:2000 concentration in blocking solution and incubated for 1 hour prior to washing. Protein detection was performed using enhanced chemiluminescence (ECL) (Thermo Scientific, Waltham, MA) according to the directions of the manufacturer. Imaging of Western blots was performed using a Bio-Rad ChemiDoc-MP imaging system. To assess protein-loading immunoblots were stripped using Restore Western Blot Stripping Buffer (Thermo Scientific, Waltham, MA) according to the directions of the manufacturer and immunoblotting was performed using mouse monoclonal antibodies to b-actin (0.5 μg/ml) in blocking solution and an HRP-linked anti-mouse secondary antibody (1:2000). Densitometry was performed using ScanAnalysis 2.5 software (Biosoft). For the analyses, the densitometric data for the protein signal were divided by the matched signal for b-actin to compare separate immunoblots, densitometric data were normalized to WT controls.

## Expression of glomerular mRNAs

Reverse transcription (RT) followed by a quantitative polymerase chain reaction (Q-RT-PCR) was performed using an iCycler ™ (Bio- Rad Laboratories, Inc., Hercules, CA). For the studies, total cellular RNA was prepared using glomerular preparations and the Trizol reagent according to the manufacturer's directions (Life Technologies Inc., Carlsbad, CA). The RNA was treated with RNAase free DNAase (Qiagen) and then reverse-transcribed with Superscript reverse transcriptase (Invitrogen) and oligo (dT) primers. Real-time quantitative PCR was performed using an iCycler [Q-PCR machine] and the universal SYBR Green PCR Master Mix Kit (Perkin-Elmer, Waltham, MA). The amplification signals were normalized to the endogenous GAPDH mRNA level. The primer sequences used for Q-RT-PCR were as follows: CNP forward: 5'- AAT ACA AAG GCG GCA ACA AG – 3' & reverse, 5' – TAA CAT CCC AGA CCG CTC AT – 3'. NPRC forward, 5' – AGC TGG CTA CAG CAA GAA GG – 3' & reverse, 5' – CGG CGA TAC CTT CAA ATG TC – 3'; GAPDH, 5'_- GTGAAGGTCGGTGTG AACG-GATTTG – 3' & antisense 5' -ACATTGGGGGTAGGAACACGGAAGG - 3'; Fibronectin forward 5' – CGA GGT GAC AGA GAC CAC AA – 3' & reverse 5' – CTG GAG TCA AGC CAG ACA CA – 3'; collagen type 1, alpha-1 (COL1A1) forward 5' – ATC TCC TGG TGC TGA TGG AC – 3' & reverse 5' – ACC TTG TTT GCC AGG TTC AC – 3'; alpha-smooth muscle actin (SMA) forward 5' – GAG GCA CCA CTG ACC CCT AA – 3' & reverse 5' –CAT CTC CAG AGT CCA GCA CA – 3'. Collagen 4, alpha 1 (COL4A1) forward 5' – TATCTCTGGGG-ACAACATCCG – 3' & reverse 5' – CATCTCGCTTCTCTCTATGGTG – 3', Nephrin forward 5' –– AGCTACCCTGCATAGCCAGA – 3' & reverse 5' – – CCCAAGCTATGGACACTGGT – 3', Collagen 4, alpha 4 (COL4A4) forward 5' – CTCCTGGTTCTCCACAGTCAG – 3' & reverse 5' – AAGGGCAGAGTGCTAAACACA – 3', Podocin forward 5' – GTGTCCAAAGC-CATCCAGTT – 3' & reverse 5' – GCAATGCTCTTCCTTTCCAG – 3'.

## Statistical analysis

Data are presented as the mean ± standard error of the mean (SEM) and statistical analyses were performed using the Prism 9 computer program (GraphPad Software, Inc.) assuming a normal distribution of the data. For comparison of two groups of continuous variables, data was analyzed by a t-test. For comparison of more than 2 groups of continuous variables, data was analyzed by either a one-way or two-way analysis of variance (ANOVA) as specified by the Prism program, followed by Sidak's multiple comparisons post-test. For non-continuous variables (histopathology), data was analyzed using a Fishers Exact test using the number mice with the specified histologic abnormality. Graphs of the histopathologic findings were, however, presented as the percentage of mice with the specified abnormality to permit a more effective comparison of the differences between the experimental groups in studies with an imbalance in the number of mice in each group. All statistics were performed using two-sided tests.

## Supporting information

**S1 Data.**
(PDF)

## Author contributions

**Conceptualization:** Robert F. Spurney.

**Data curation:** Robert F. Spurney, Liming Wang.

**Formal analysis:** Robert F. Spurney, Liming Wang.

**Funding acquisition:** Robert F. Spurney.

**Investigation:** Robert F. Spurney, Liming Wang, Yuping Tang, Anne F. Buckley.

**Methodology:** Robert F. Spurney, Anne F. Buckley.

**Project administration:** Robert F. Spurney.

**Resources:** Robert F. Spurney.

**Supervision:** Robert F. Spurney, Liming Wang.

**Validation:** Robert F. Spurney.

**Writing – original draft:** Robert F. Spurney, Liming Wang.

**Writing – review & editing:** Robert F. Spurney, Liming Wang.

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
