## [Decision Letter · Decision Letter 0]

10 Sep 2024

PONE-D-24-36618Podocyte specific knockout of the natriuretic peptide clearance receptor is podocyte protective in focal segmental glomerulosclerosisPLOS ONE

Dear Dr. Spurney,

Thank you for submitting your manuscript to PLOS ONE. After careful consideration, we feel that it has merit but does not fully meet PLOS ONE’s publication criteria as it currently stands. Therefore, we invite you to submit a revised version of the manuscript that addresses the points raised during the review process. Please submit your revised manuscript by Oct 25 2024 11:59PM. If you will need more time than this to complete your revisions, please reply to this message or contact the journal office at plosone@plos.org . Please include the following items when submitting your revised manuscript:

We look forward to receiving your revised manuscript.

Kind regards,

Michael Bader

Academic Editor

PLOS ONE

https://pubmed.ncbi.nlm.nih.gov/34755480/

In your revision ensure you cite all your sources (including your own works), and quote or rephrase any duplicated text outside the methods section. Further consideration is dependent on these concerns being addressed.

4. We note that the grant information you provided in the ‘Funding Information’ and ‘Financial Disclosure’ sections do not match. When you resubmit, please ensure that you provide the correct grant numbers for the awards you received for your study in the ‘Funding Information’ section.

“1. R21 TR004257-01 from the National Institutes of Health.

2.  BX002984 from the Veterans Administration Merit Review Program.

3.  W81XWH-19-1-0314 from the United States Department of Defense.”

6. We note that your Data Availability Statement is currently as follows: All relevant data are within the manuscript and its Supporting Information files.

Please confirm at this time whether or not your submission contains all raw data required to replicate the results of your study. Authors must share the “minimal data set” for their submission. PLOS defines the minimal data set to consist of the data required to replicate all study findings reported in the article, as well as related metadata and methods (https://journals.plos.org/plosone/s/data-availability#loc-minimal-data-set-definition). For example, authors should submit the following data: - The values behind the means, standard deviations and other measures reported; - The values used to build graphs; - The points extracted from images for analysis. Authors do not need to submit their entire data set if only a portion of the data was used in the reported study. If your submission does not contain these data, please either upload them as Supporting Information files or deposit them to a stable, public repository and provide us with the relevant URLs, DOIs, or accession numbers. For a list of recommended repositories, please see https://journals.plos.org/plosone/s/recommended-repositories. If there are ethical or legal restrictions on sharing a de-identified data set, please explain them in detail (e.g., data contain potentially sensitive information, data are owned by a third-party organization, etc.) and who has imposed them (e.g., an ethics committee). Please also provide contact information for a data access committee, ethics committee, or other institutional body to which data requests may be sent. If data are owned by a third party, please indicate how others may request data access.

7. When completing the data availability statement of the submission form, you indicated that you will make your data available on acceptance. We strongly recommend all authors decide on a data sharing plan before acceptance, as the process can be lengthy and hold up publication timelines. Please note that, though access restrictions are acceptable now, your entire data will need to be made freely accessible if your manuscript is accepted for publication. This policy applies to all data except where public deposition would breach compliance with the protocol approved by your research ethics board. If you are unable to adhere to our open data policy, please kindly revise your statement to explain your reasoning and we will seek the editor's input on an exemption. Please be assured that, once you have provided your new statement, the assessment of your exemption will not hold up the peer review process.

8. Please include your full ethics statement in the ‘Methods’ section of your manuscript file. In your statement, please include the full name of the IRB or ethics committee who approved or waived your study, as well as whether or not you obtained informed written or verbal consent. If consent was waived for your study, please include this information in your statement as well.

10. PLOS ONE now requires that authors provide the original uncropped and unadjusted images underlying all blot or gel results reported in a submission’s figures or Supporting Information files. This policy and the journal’s other requirements for blot/gel reporting and figure preparation are described in detail at https://journals.plos.org/plosone/s/figures#loc-blot-and-gel-reporting-requirements and https://journals.plos.org/plosone/s/figures#loc-preparing-figures-from-image-files. When you submit your revised manuscript, please ensure that your figures adhere fully to these guidelines and provide the original underlying images for all blot or gel data reported in your submission. See the following link for instructions on providing the original image data: https://journals.plos.org/plosone/s/figures#loc-original-images-for-blots-and-gels. In your cover letter, please note whether your blot/gel image data are in Supporting Information or posted at a public data repository, provide the repository URL if relevant, and provide specific details as to which raw blot/gel images, if any, are not available. Email us at plosone@plos.org if you have any questions.

Reviewers' comments:

Reviewer's Responses to Questions

**Comments to the Author**

1. Is the manuscript technically sound, and do the data support the conclusions?

Reviewer #1: Partly

Reviewer #2: Partly

2. Has the statistical analysis been performed appropriately and rigorously? 

Reviewer #1: Yes

Reviewer #2: Yes

3. Have the authors made all data underlying the findings in their manuscript fully available?

Reviewer #1: Yes

Reviewer #2: Yes

4. Is the manuscript presented in an intelligible fashion and written in standard English?

Reviewer #1: Yes

Reviewer #2: Yes

5. Review Comments to the Author

Reviewer #1: Comments to “Podocyte specific knockout of the natriuretic peptide clearance receptor is podocyte protective in focal segmental glomerulosclerosis” from Liming Wang, Yuping Tang, Anne F. Buckley and Robert F. Spurney

This experimental study addresses the original and clinically relevant question whether deletion (KO) of the natriuretic peptide receptor C (NPR-C) in podocytes augments the renoprotective effects of natriuretic peptides. To this aim, the authors generated mice with deletion of NPRC in podocytes and studied them in an experimental model of glomerulosclerosis. The study focusses on C-type natriuretic peptide (CNP). An explanation why ANP and BNP were not considered as mediators of the protective effects of NPRC KO is not provided.

The results are potentially interesting but the quality of the experiments, including the quality of the illustrating figures, are too low to evaluate validity (specific criticisms are listed below). This especially concerns the illustrations of immunohistochemistry (stainings are diffuse, negative controls and scale bars are missing). Mechanistic insights to the antiapoptotic and antifibrotic effects of podocyte NPRC deletion are not provided. Moreover, the authors did not consider the possibility that the “protection” of NPRC-deficient podocytes was partly mediated by inhibition of the Gi/Gq-coupling functions of this receptor (doi: 10.1038/s41392-023-01560-y; and many other references). In other words, a clear demonstration whether NPRC KO was protective due to augmented NP/NPRB (or NPRA)/cGMP signalling and/or due to diminished NPRC/G-protein signalling is missing. This limitation must be addressed or at least carefully discussed.

Specific Comments:

- Introduction (page 3): “1. Podocytes express NPRC, NPRB and NPRC [13-17]”. Please correct the redundancy in this sentence.

- Nomenclature: for many good reasons, the IUPHAR recommended to name NPRA and NPRB as GC-A and GC-B. At least you should clarify this nomenclature.

- Introduction (page 4): “1. NPs potently stimulated cGMP generation and inhibited podocyte apoptosis in cultured podocytes[17], and 2. Pharmacological blockade of NPs in vivo enhanced urinary cGMP excretion and significantly reduced albuminuria in a mouse model of FSGS[17].” These two sentences obviously contradict each other. Please correct them or discuss appropriately.

- Throughout the manuscript there are numerous grammatical mistakes and incomplete sentences, which leaves the reviewer with the impression of quick superficial writings (for instance on pages 1 and 2 of results section: detecteded, anibody, exression, nomral, podocytres, …)

- Figure 1:

o The quality of the images in Figure 1A is very low and must be markedly improved.

o Figures 1B and C illustrate “statements” (CNP is produced by podocytes and binds to…; NPRC clears CNP from the circulation) which were not experimentally addressed by the present studies. If these statements/schemes are based on published data, then please refer those in the legend. As far as this reviewer knows, nobody has previously shown that “NPRC clears CNP from the circulation”).

o Figure 1E: How is the expression of NPRB and NPRC in these cells? Is it correct that CNP barely increased cGMP contents of cultured control podocytes but raised the cGMP contents of NPR-C KO podocytes by more than 50-fold? This is a surprisingly huge difference. It suggests that in presence of NPR-C, CNP/NPRB/cGMP signalling is almost fully blocked. Why? As compared: how are the cGMP responses of control and KD podocytes to ANP and/or BNP?

o Figure 1G: results are unclear and possibly incomplete. Please show the % apoptosis in control and KO cells in Serum and Serum-free medium. Is there a difference in serum-containing medium between genotypes? The increased apoptosis is interesting and should be confirmed with additional techniques and complemented with mechanistic insights.

- Figure 2:

o How is the protein level of NPR-C in glomeruli from both genotypes? Is NPR-C expression preserved in other types of cells? (was NPR-C expression in rest of kidney, after separation of glomeruli, preserved?)

- Figure 4:

o Figure 4C: The y-axis is named number of mice with Glomerulosclerosis. The legend shows normal, mild, moderate and severe sclerosis. Please label the y-axis accordingly (staging of glomerulosclerosis, histology of glomeruli?). It would help understandings to present the data in %, as is described in corresponding results section. (“~86% of the WT controls have no glomerulosclerosis and ~14 % mild glomerulosclerosis”). The present Figure is totally unclear because it does not relate the “depicted numbers” to the total number of study mice.

o Figure 4D-F: The columns “Absent” and “Present” in the diagram are redundant. Please show only the column “Present”. Why did you interchange black/white in Fig. E?

o Does podocyte apoptosis induce tubulointerstitial inflammation? Where is the link between these two observations? Figure 4E must illustrate in clear way how inflammation was measured.

- Figure 5:

o The mRNA and protein results are very discrepant. The mRNA expression levels of alpha-SMA and Fibronectin are only slightly impacted by genotypes and conditions (Figure 5A-D). The Western blots instead show huge inductions in WT mice and almost full inhibition of this induction in the KO (Figures 5E-H). How do you explain this mismatch between the regulation of mRNA and protein levels?

- Table S2:

o As mentioned in the discussion, NPs inhibit the master regulator of fibrosis, TGF-beta, by phosphorylating SMADs on an inhibitory site. Did you analyse SMAD phosphorylation? Is the inhibitory phosphorylation increased in the KO mice? Do NPRC-deficient podocytes express less proinflammatory and profibrotic molecules?

- Please discuss appropriately the discrepant results from Figures 3 (marked Albuminuria) and 7 (Cystatin C levels barely affected)

- Methods: It is difficult to understand how the diverse transgenic mouse models were generated. It could help to include a scheme of the mice breedings.

Reviewer #2: The present work deals with the role of the natriuretic peptide receptor C (NPRC), which has a functional significance as a clearance receptor for the natriuretic peptides ANP, BNP and CNP, in podocytes. In the first part of the paper, the authors show that cultured murine podocytes express CNP and that knockdown of NPRC is associated with higher CNP levels in the medium and attenuated apoptosis induced by serum deprivation. The authors conclude that blockade or inactivation of NPRC in podocytes may have a protective effect in glomerular injury models. They therefore exposed mice with podocyte-specific deletion of NPRC to a corresponding damage model (FSGS model). Indeed, some glomerular and interstitial damage markers are ameliorated in NPRC KO mice, supporting the authors' hypothesis.

The study is the logical continuation of earlier studies by the authors with pharmacological blockade of the NPRC. However, there are some issues to consider.

1. The two parts of the study, in vitro and in vivo, are linked by the fact that the NPRC was genetically inactivated in both. Otherwise, the results of the in vitro studies are not taken up in the in vivo part. Does the local production of CNP in podocytes play also a role in vivo in terms of an autocrine/paracrine effect? Is CNP regulated in podocytes in the animal model used? Or are the observed protective effects of NPRC deletion due to an enhanced effect of ANP / BNP via the NPRA? It is understandable that this question is very difficult to answer experimentally, but it should at least be discussed in more detail.

2. Line 6, results: Reference no. 23 is given as a reference for the expression of NPRB in podocytes. This should be checked, as the paper mentioned deals with the expression of CNP in the kidney, but not with NPRB. Other studies cited as evidence for the expression of NPRB in podocytes also show very strong expression of NPRA (receptor for ANP and BNP) and NPRC, but at best very weak expression of NPRB in podocytes (references 14 and 16). On the contrary, a recent study using in situ hybridization showed the expression of NPRB in glomerular endothelial cells and mesangial cells, but not in podocytes (Heinl, ES et al.; Pflugers Archiv 2023, 475(3): 343-360). Taking into account the results of these studies, in particular the strongly dominant expression of NPRA in podocytes, the influence of NPRC knockdown on the effects of ANP or BNP in cultured podocytes (cGMP, apoptosis) should be investigated.

3. Fig. 1A: CNP immunofluorescence staining, at least in the present PDF, is not of sufficient quality to detect co-staining in podocytes. The quality of the image should be improved. It is also noted that "CNP was widely expressed in multiple cell types in the kidney". Please show overview images that reflect this expression.

4. Fig. 1E: It is clearly shown that the effect of CNP on cGMP levels is significantly enhanced in NPRC knockdown cells compared to control cells. It is not clear from Fig. 1E whether CNP has a significant effect at all in control cells. Does cGMP increase in control cells under CNP or is the effect completely abolished in the presence of NPRC.

5. The results of the in vitro experiments are compatible with the conclusion that the endogenous production of CNP in podocytes has a protective effect and that this is mediated via the NPRB. To further substantiate this, the NPRB could be knocked down in the podocyte cell culture used, with and without simultaneous NPRC inactivation, and the effect on apoptosis investigated. Overall, it should at least be discussed that there are a variety of studies showing that the NPRC is not only a clearance receptor, but also a signaling receptor that can be activated by CNP (for example: Bubb KJ, et. al.; Circulation. 2019 ;139(13): 1612-1628.; Moyes AJ, Hobbs AJ. Int J Mol Sci. 2019 ;20(9): 2281).

6. Fig. 4: Please show absolute values and not percentages.

7. Fig. 5G: The arrow for fibronectin does not point to the dominant band of the Western blot. Which is the correct band? The molecular weight or a standard should be shown.

8. First page of introduction, last line: It should read "NPRA, NPRB and NPRC" instead of "NPRC, NPRB and NPRC".

6. PLOS authors have the option to publish the peer review history of their article (what does this mean? ). If published, this will include your full peer review and any attached files.

**Do you want your identity to be public for this peer review?** For information about this choice, including consent withdrawal, please see our Privacy Policy .

Reviewer #1: No

Reviewer #2: No

---

## [Author Response · Author response to Decision Letter 0]

21 Nov 2024

Reviewer #1:

1. Histologic studies: Immunohistochemistry staining is diffuse and lacks negative controls.

We had significant difficulty locating antibodies for the immunohistochemistry studies, and were unable to locate antibodies for mouse kidney. As a result, we attempted to stain human kidney specimens with several antibodies and the current figure was the result. The antibody we used stained multiple cell types in the kidney the kidney consistent with published single cell transcriptomic studies (for example see reference [1]) Little staining was detected in the isotype control antibody panels. We agree that the images a suboptimal and it is also possible that the diffuse staining in the tope panels of Figure 1A could be non-specific. As a result, we are removing Figure 1A from the manuscript.

2. Scale bars are missing:

We have added scale bars to the current histology pictures figures.

3. Alternative considerations: Did not consider that the “protection” of NPRC-deficient podocytes was partly mediated by inhibition of the Gi/Gq-coupling functions of this receptor.

We are aware of alternative signaling pathways linked to NPRC and we discussed this possibility in a previous publication examining the effects of pharmacologic NPRC blockade in a mouse model of FSGS [2]. We apologize for not include this possibility in the current publication and we have amended the manuscript to address this possibility in the revised manuscript.

4. Nomenclature: for many good reasons, the IUPHAR recommended to name NPRA and NPRB as GC-A and GC-B. At least you should clarify this nomenclature.

We have acknowledged the alternate designations for NPRA and NPRB in the text of the manuscript.

5 Introduction: Contradictory sentences: “1. NPs potently stimulated cGMP generation and inhibited podocyte apoptosis in cultured podocytes[17], and 2. Pharmacological blockade of NPs in vivo enhanced urinary cGMP excretion and significantly reduced albuminuria in a mouse model of FSGS[17].” These two sentences obviously contradict each other. Please correct them or discuss appropriately.

We apologize for the error, and we have corrected the mistake as outlined below.

The corrected statement is shown below:

NPs potently stimulated cGMP generation and inhibited podocyte apoptosis in cultured podocytes[17], and 2. Pharmacological blockade of NPRC in vivo enhanced urinary cGMP excretion and significantly reduced albuminuria in a mouse model of FSGS[17].” These two sentences obviously contradict each other. Please correct them or discuss appropriately.

6. Figure 1A: The quality of the images in Figure 1A is very low and must be markedly improved.

We agree that the Immunohistochemistry studies are suboptimal. See response #1 above.

6. Figures 1B and C illustrate “statements” (CNP is produced by podocytes and binds to…; NPRC clears CNP from the circulation) which were not experimentally addressed by the present studies. If these statements/schemes are based on published data, then please refer those in the legend. As far as this reviewer knows, nobody has previously shown that “NPRC clears CNP from the circulation”).

The natriuretic peptide (NP) clearance receptor (NPRC) binds ANP, BNP, and CNP with a high affinity in the low picomolar range [3-6] and either pharmacologic blockade or KO of NPR-C suggests that a major function of NPRC is to clear NPs from the circulation [7, 8]. The negative regulatory effects of NPRC on the functions of CNP have been demonstrated in bone using mouse models. In this regard, mutations that disrupt the function of genes encoding CNP or NPRB (CNP receptor) causes dwarfism [9-11]. In contrast, disruption of the gene encoding the clearance receptor (NPRC) cause bone overgrowth [8]. These data suggest that: 1. A major function of CNP is binding to NPRB and stimulating bone growth, and 2. the stimulatory effects of CNP on bone growth are augmented by decreased clearance of CNP from the circulation by NPRC.

7. How is the expression of NPRB and NPRC in these cells?

We have provided data on expression of NPRA, NPRB and NPRC in Figure 2. The major findings were: 1. NPRC expression was effectively decreased in KD cells, 2. NPRC is robustly expressed in podocytes compared to either NPRA or NPRB, and 3. NPRB expression is downregulated in NPRC KO mice.

8. Is it correct that CNP barely increased cGMP contents of cultured control podocytes but raised the cGMP contents of NPR-C KO podocytes by more than 50-fold? This is a surprisingly huge difference. It suggests that in presence of NPR-C, CNP/NPRB/cGMP signaling is almost fully blocked. Why?

It is correct that CNP induced cGMP generation is significantly blunted by the high expression levels of both NPRC [1, 12, 13] and proteolytic cleavage of NPs by neprilysin [2, 14, 15]. Moreover, the 0.01 nM concentration of CNP is at the lower limit of NPRB binding affinity [5, 6, 16], which severly limits the response to CNP at the 0.01 nM concentration (Figure 1E). As a result, calculating the fold increase in cGMP generation using a number close to zero results in a very large fold change. In contrast, the higher concentrations of CNP are more effectively bound by NPRB (GC-B) and results in more robust cGMP response in, although the response is likely tempered by the by both the clearance receptor (NPRC) and peptide cleavage by neprilysin (both are expressed in podocytes [2]).

9. Figure 1E: As compared: how are the cGMP responses of control and KD podocytes to , and/or BNP?

We previously measured cGMP generation using 1µM concentrations of ANP and CMP. In these studies, ANP stimulated cAMP to a greater extent than CNP but the difference was not statistically different [2]. In the current study (Figure 2A), we saw the same pattern. ANP stimulated a greater increase in cGMP generation compared to CNP, but the differences were not statistically different at the 1.0 nM or 10 nM concentrations. In addition, the same pattern was observed in NPRC KD cells, although cGMP response was enhanced compared to cells expressing NPRC. See Figure 2A for current results.

10. Figure 1G: Results are unclear and possibly incomplete. Please show the % apoptosis in control and KO cells in Serum and Serum-free medium. Is there a difference in serum-containing medium between genotypes?

As described in the initial submission, we combined serum treated control and KD podocyte data because little apoptosis was observed in both groups. In the revised manuscript, we have repeated the experiments and separated both the serum and serum free data for KD and control podocytes. In addition, we have increased the duration of the serum deprivation to enhance the apoptotic effect to increase the apoptotic response and, in turn, provide a greater range to enhance sensitivity for detecting differences. In the current study (Figure 1F), KD of NPRC reduced apoptosis induced by serum deprivation, and this beneficial effect of NPRC KD was partially attenuated by pharmacologic blockade of NPRB with pharmacologic blockade pf NPRB with P19 [17, 18]. There was no statistically significant difference in apoptosis between control and KD cells treated with serum. These data are consistent with the notion that beneficial effects of NPRC KO on the apoptotic response was mediated, at least part, by activation of NPRB by CNP.

11. Figure 1G: The increased apoptosis is interesting and should be confirmed with additional techniques and complemented with mechanistic insight.

Serum deprivation has been utilized for several decades to induce cell death by the removal of pro-survival growth factors [19]. Moreover, our lab and others have used serum deprivation to induce apoptosis in podocytes [2, 20]. In response to your comment, the mechanisms causing cell death has expanded to include multiple distinct pathways [21]. In addition, there is overlap between of the pathways promoting cell death. For example, autophagy often progresses to apoptosis [21, 22] and if apoptotic cells are not efficiently cleared, the cells undergo programed cell necrosis [21-23]. The methodology used in our study (annexin V staining) is commonly used to detect apoptosis and, given the complexity of investigating types of cells death and the current focus of this manuscript, we believe the studies are beyond the focus of this manuscript.

12. Figure 2: How is the protein level of NPR-C in glomeruli from both genotypes? Is NPR-C expression preserved in other types of cells? (was NPR-C expression in rest of kidney, after separation of glomeruli, preserved?)

As discussed in the methods section, the Cre mouse was obtained from Jackson labs (model No. 008205) and permits inducible expression of Cre recombinase specifically in podocytes. The podocyte specificity of the system was documented in the initial publication [24] and this model has been used in multiple manuscripts since the initial paper was published. Figure 3A&B and supplementary figure S2 demonstrate that expression of Cre recombinase under the regulation of the podocyte specific podocin promoter in tdTomato mice (see methods) causes expression of green fluorescence in a pattern consistent with expression of Cre recombinase specifically in podocytes. In mice containing 2 “floxed” alleles of NPRC, this same promoter system specifically deletes NPRC in podocytes. In addition, we demonstrated that levels of NPRC were significantly decreased in the glomerular preparations by both quantitative RT-PCR (figure 2C) and immunoblotting (figure 2D & 2E) despite expression of NPRC in multiple cell types in the kidney [1, 12, 25]. Taken together with a large literature suggesting specificity of this approach to KO of podocyte proteins, these data suggest that podocyte specific KO of NPRC was successful.

13. Figure 4: The y-axis is named number of mice with Glomerulosclerosis. The legend shows normal, mild, moderate and severe sclerosis. Please label the y-axis accordingly (staging of glomerulosclerosis, histology of glomeruli?). It would help understandings to present the data in %, as is described in corresponding results section. (“~86% of the WT controls have no glomerulosclerosis and ~14 % mild glomerulosclerosis”). The present Figure is totally unclear because it does not relate the “depicted numbers” to the total number of study mice. Figure 4D-F: The columns “Absent” and “Present” in the diagram are redundant. Please show only the column “Present”.

The percentage of mice with glomerulosclerosis (GS) relates to number of mice with (absent) or without (Present) evidence of GS. For example, in Figure 4C, the TG wild type group has 3 mice without GS, 5 mice with mild GS, 2 mice with moderate GS and 11 mice with severe GS. The total number of mice is 21. As a result, there are 18 mice with GS (5+2+11) divided by 21 mice without GS, which is approximately ~85.7% of TG wild type mice with GS.

To simplify Figure 4, we have eliminated Figure 4C from the manuscript and reported the percentage of mice with or without GS; however, we have retained Figure 4C in the Supporting Data to provide the exact numbers of mice with mild, moderate and severe GS as reported by our pathologist (Dr. Buckley). We decided to present the graphs of the histopathologic findings as the percentage of mice with the specified abnormality to permit a more effective comparison of the differences between the experimental groups in studies with an imbalance in the number of mice in each group; however, the statistical evaluation was performed using the number of mice with the specified histologic abnormality.

14. Figure 4E: Why did you interchange black/white in Fig. E?

We have corrected the black/white problem in labeling the percentages.

15. Figure 4E: Does podocyte apoptosis induce tubulointerstitial inflammation? Where is the link between these two observations?

Podocyte damage/cell death disrupts the glomerular filtration barrier and promotes high levels of proteinuria. As discussed in the text of the manuscript, the level of proteinuria may contribute to tubulointerstitial inflammation. Filtered proteins are resorbed by the proximal tubules and processed by lysosomes and endoplasmic reticulum [26]. In the setting of heavy proteinuria, these intracellular pathways are stressed [27], which results in the secretion of cytokines that attract and activate inflammatory cells [28] and produce profibrotic mediators such as TGF-beta [29].

16. Figure4E: Figure 4E must illustrate in clear way how inflammation was measured.

The pathologist evaluating the renal histopathology provided a detailed descript of the methodology for assessing tubulointerstitial inflammation. Please see the Methods Section for details.

17. Figure 5: The mRNA and protein results are very discrepant. The mRNA expression levels of alpha- SMA and Fibronectin are only slightly impacted by genotypes and conditions (Figure 5A-D). The Western blots instead show huge inductions in WT mice and almost full inhibition of this induction in the KO (Figures 5E-H). How do you explain this mismatch between the regulation of mRNA and protein levels?

We also noticed the discrepancy in the mRNA expression and protein levels in the glomerular preparations. Review of the literature suggests that fibronectin accumulates in the glomerulus through both local synthesis and by deposition from the circulation [30, 31]. This was demonstrated by a liver specific fibronectin knockout to decrease circulating fibronectin [31]. In these KO mice, there is a decrease in both mesangial expansion and fibronectin accumulation the kidneys of diabetic mice. In addition, the major source of fibronectin in early stages of FSGS is from the circulation with deposition of glomerular fibronectin occurring later in the disease process. We suspect that the discrepancy in the mRNA expression and protein levels is related to greater deposition of fibronectin from the circulation in damaged glomeruli compared to glomeruli with less severe glomerular injury. This possibility has been mentioned in the discussion section.

18. As mentioned in the discussion, NPs inhibit the master regulator of fibrosis, TGF-beta, by phosphorylating SMADs on an inhibitory site. Did you analyze SMAD phosphorylation? Is the inhibitory phosphorylation increased in the KO mice? Do NPRC-deficient podocytes express less proinflammatory and profibrotic molecules?

TGF-beta activates its receptor, which phosphorylates Smad3 on Ser423 and Ser425 [32].. Phosphorylated Smad3 combines with Smad4 to form a Smad complex,which then translocates into the nucleus to initiate mRNA transcription associate with inflammation and collagen formation [32].. ANP stimulates cGMP generation and actives cGMP kinase 1 (cGK1), which then  phosphorylates Smad3 on Ser309 and on Thr388, and inhibits the nuclear translocation of the Smad complex and , in turn, fibrosis [32]. 00042.2016). Despite an extensive search, we were unable to locate antibodies to phospho-SMAD3 Ser309 or phospho-SMAD3 Thr388.

There are antibodies for phosphor-Smad3 on Ser423 and Ser425, and we attempted to determine if these Smad phosphoryaltion sites were affected in our KO mice but were unable to detect a signal. It is difficult to interpret these results because exploring cell signaling in animal models is complicated by: 1. Changes in modifications to cell signaling molecules involved in the response (for example phosphorylation and dephosphorylation in vivo, and repeated freeze-thaw cycles of stored samples ) and 2. The time point chosen to harvest tissues (late versus early in the fibrotic response). We did not screen for additional proinflammatory or profibrotic molecules.

20. Proteinuria versus kidney function: Please discuss appropriately the discrepant results from Figures 3 (marked Albuminuria) and 7 Cystatin C levels barely affected)

It is common to have a large disparity in proteinuria and renal function in both animals and humans, especially in the early stages of glomerular diseases. The amount of proteinuria is dependent on the integrity of the filtration barrier, and podocytes play a key role in maintaining permselectivity. In contras

---

## [Decision Letter · Decision Letter 1]

6 Dec 2024

PONE-D-24-36618R1Podocyte specific knockout of the natriuretic peptide clearance receptor is podocyte protective in focal segmental glomerulosclerosisPLOS ONE

Dear Dr. Spurney,

Thank you for submitting your manuscript to PLOS ONE. After careful consideration, we feel that it has merit but does not fully meet PLOS ONE’s publication criteria as it currently stands. Therefore, we invite you to submit a revised version of the manuscript that addresses the points raised during the review process.

We look forward to receiving your revised manuscript.

Kind regards,

Michael Bader

Academic Editor

PLOS ONE

Journal Requirements:

Reviewers' comments:

Reviewer's Responses to Questions

**Comments to the Author**

1. If the authors have adequately addressed your comments raised in a previous round of review and you feel that this manuscript is now acceptable for publication, you may indicate that here to bypass the “Comments to the Author” section, enter your conflict of interest statement in the “Confidential to Editor” section, and submit your "Accept" recommendation.

Reviewer #2: (No Response)

2. Is the manuscript technically sound, and do the data support the conclusions?

Reviewer #2: Partly

3. Has the statistical analysis been performed appropriately and rigorously? 

Reviewer #2: Yes

4. Have the authors made all data underlying the findings in their manuscript fully available?

Reviewer #2: Yes

5. Is the manuscript presented in an intelligible fashion and written in standard English?

Reviewer #2: Yes

6. Review Comments to the Author

Reviewer #2: The authors have addressed most of my criticisms and suggestions and have significantly improved the manuscript. However, there are still points that need to be clarified:

1. As suggested, experiments were carried out on immortalized podocytes to examine the relevance of CNP signaling via NPRB. For this purpose, the substance P19 was used (Figure 1G). The effect of P19 in KD cells is low and the significance is very limited with a test number of n=4. This could easily be corrected by additional experiments. Furthermore, the concentration of P19 used is not indicated in the manuscript. This is particularly important because the affinity of P19 for NPRA and NPRB does not differ greatly, and the greatest affinity is actually for NPRC. This information can be found in reference 26 and should at least be discussed with all its consequences. A knockdown of NPRB would be the better method here.

2. The experiments comparing the effects of ANP and CNP on cGMP and apoptosis are very helpful. They clearly show that ANP exerts a protective effect on podocytes even without simultaneous inactivation of NPRC. This fits very well with in vivo experiments by other groups. However, the finding that a protective effect of CNP is only present when NPRC is inactivated calls into question the in vivo relevance of this effect, since podocytes in vivo also express NPRC. This should be discussed.

3. As in the first version of the manuscript, the references should be carefully checked. For example, references 14 and 16 are mentioned to prove that “CNP is widely expressed in multiple cell types in mouse and human kidneys including glomerular podocytes (results, first paragraph)”. This information cannot be found in the papers. There may be a misunderstanding because “CNP” is mentioned in the gene lists. However, this is the gene of a “cyclic nucleotide phoshodiesterase” and not CNP whose gene name is NPPC. In addition, the references are used as evidence for “for the formation of cGMP in podocytes by the NPRB” (results, first paragraph). This is also not correct. Please correct me if I am wrong.

4. It should be briefly discussed that the blood pressure values show a very high variability. Although this can occasionally occur with automated tail cuff measurement and is therefore not very unusual, it significantly weakens the authors' conclusion that there are no blood pressure differences between the genotypes.

5. Legend Figure 2B: “KO mice” should be corrected to “KD cells”

7. PLOS authors have the option to publish the peer review history of their article (what does this mean? ). If published, this will include your full peer review and any attached files.

**Do you want your identity to be public for this peer review?** For information about this choice, including consent withdrawal, please see our Privacy Policy .

Reviewer #2: No

---

## [Author Response · Author response to Decision Letter 1]

13 Jan 2025

RESPONSE TO THE REVIEW

Comments to the authors:

1. If the authors have adequately addressed your comments raised in a previous round of review and you feel that this manuscript is now acceptable for publication, you may indicate that here to bypass the “Comments to the Author” section, enter your conflict of interest statement in the “Confidential to Editor” section, and submit your "Accept" recommendation.

- Reviewer #2: No response

- We have responded to individual comments below.

2. Is the manuscript technically sound, and do the data support the conclusions? The manuscript must describe a technically sound piece of scientific research with data that supports the conclusions. Experiments must have been conducted rigorously, with appropriate controls, replication, and sample sizes. The conclusions must be drawn appropriately based on the data presented.

- Reviewer #2: Partly

- We have made changes in the manuscript as outlined below to address comments raised by the reviewers.

3. Has the statistical analysis been performed appropriately and rigorously?

- Reviewer #2: Yes.

4. Have the authors made all data underlying the findings in their manuscript fully available?

- Reviewer #2: Yes

5. Is the manuscript presented in an intelligible fashion and written in standard English?

- Reviewer #2: Yes

Reviewer comments to the authors:

1. As suggested, experiments were carried out on immortalized podocytes to examine the relevance of CNP signaling via NPRB. For this purpose, the substance P19 was used (Figure 1G). The effect of P19 in KD cells is low and the significance is very limited with a test number of n=4. This could easily be corrected by additional experiments. Furthermore, the concentration of P19 used is not indicated in the manuscript. This is particularly important because the affinity of P19 for NPRA and NPRB does not differ greatly, and the greatest affinity is actually for NPRC. This information can be found in reference 26 and should at least be discussed with all its consequences. A knockdown of NPRB would be the better method here.

We have performed the experiments recommended by the reviewers and, although we only used an “N” of 4, the data is statistically significant, even after correcting for multiple comparisons. We do apologize for not including the concentrations of P19 (250 nM) in the manuscript, and this data has been added to the Methods Section. As the reviewers mentioned, the affinity of P19 for NPRA and NPRB does not differ greatly, and the greatest affinity is actually for NPRC. This is an important consideration that we did not address in the revised manuscript. Based on the following reference (Peptides 26 (2005) 517–524), we suggest that binding of P19 to either NPRC or NPRA is unlikely to affect the results in the knockdown (KD) podocytes because: 1. NPRC is absent in the KD cells, 2. The dosage of P19 used in the studies (250 nM) effectively binds NPRB (Kd 15.4 nM) but binds minimally to NPRA (NPRA Kd 575 nM). In contrast to the KD cells, NPRC is expressed in the control cells, which could affect the apoptotic response by blocking natriuretic clearance and enhancing CNP levels. We posit that blockade of NPRC in the control cells by P19 (Fig. 1G) had a minimal effect on the results because P19 has high affinity for both NPRB (Kd ~15 nM) and NPRC (0.0134 nM) and, in turn, would effectively inhibit both NP receptors (NPRB and NPRC). In this scenario, the benefits of increasing CNP levels by blockade NPRC would be reduced by pharmacologic blockade of NPRB. In the current study, the net effect was no change in the apoptotic response (Fig. 1G).

NOTE: affinity in the P19 reference is expressed as pK units and can be converted to Kd as follows pK = −log KD).

2. The experiments comparing the effects of ANP and CNP on cGMP and apoptosis are very helpful. They clearly show that ANP exerts a protective effect on podocytes even without simultaneous inactivation of NPRC. This fits very well with in vivo experiments by other groups. However, the finding that a protective effect of CNP is only present when NPRC is inactivated calls into question the in vivo relevance of this effect, since podocytes in vivo also express NPRC. This should be discussed.

We agree that, under the conditions used in the experiments (cell culture), ANP is more effect than CNP at inhibiting podocyte apoptosis at equimolar concentrations of ANP and CNP (Figure 2D). Indeed, the podocyte protective actions of CNP are only seen in the NPRC KD cells. The current studies do not, however, address potential CNP secretion under in vivo conditions or changes in CNP secretion during stress. Moreover, there are multiple CNP secreting cells in kidney that may contribute to enhance CNP levels and “overcome” the negative regulatory effects of NPRC. However, based on the data presented in the manuscript, we have modified the statements about the autocrine protective actions of CNP in the text of the manuscript to acknowledge the lack of a podocyte protective effect in the cells expressing NPRC

3. As in the first version of the manuscript, the references should be carefully checked. For example, references 14 and 16 are mentioned to prove that “CNP is widely expressed in multiple cell types in mouse and human kidneys including glomerular podocytes (results, first paragraph)”. This information cannot be found in the papers. There may be a misunderstanding because “CNP” is mentioned in the gene lists. However, this is the gene of a “cyclic nucleotide phoshodiesterase” and not CNP whose gene name is NPPC. In addition, the references are used as evidence for “for the formation of cGMP in podocytes by the NPRB” (results, first paragraph). This is also not correct. Please correct me if I am wrong.

The references 14 and 16 are single cell RNA sequencing studies. To obtain results for specific mRNAs investigators need to query the transcriptomics databases that accompany the manuscripts.

Database for these results, are available at the following websites: 1. http://humphreyslab.com/SingleCell/ and 2. https://susztaklab.com/EH/park/. I have also added these weblinks to each manuscript citing single cell RNA sequencing studies in the Reference Section. I am also requesting guidance how to reference these results in the manuscript.

4. It should be briefly discussed that the blood pressure values show a very high variability. Although this can occasionally occur with automated tail cuff measurement and is therefore not very unusual, it significantly weakens the authors' conclusion that there are no blood pressure differences between the genotypes.

The tail cuff blood pressure measurement did show variability in the results, which reduces chances of detecting a difference between groups. We can only speculate on the causes, but the major finding in the BP studies is the lack of a difference in the Gq mice compared to the Gq-KO mice. This is important because a difference in BP could affect the severity of the kidney disease. The comparison in these 2 groups (GQ and Gq-KO) found very similar BP levels suggesting that a difference in BP did not affect the in vivo findings in the study.

5. Legend Figure 2B: “KO mice” should be corrected to “KD cells”

We have corrected this mistake in the text of the manuscript.

Additional issues: PLOS authors have the option to publish the peer review history of their article (what does this mean?). If published, this will include your full peer review and any attached files.

I do not think is necessary to publish the full review process.

References

1. Kubiak-Wlekly A, Perkowska-Ptasinska A, Olejniczak P, Rochowiak A, Kaczmarek E, Durlik M, et al. The Comparison of the Podocyte Expression of Synaptopodin, CR1 and Neprilysin in Human Glomerulonephritis: Could the Expression of CR1 be Clinically Relevant? Int J Biomed Sci. 2009;5(1):28-36. PubMed PMID: 23675111; PubMed Central PMCID: PMCPMC3614758.

2. Debiec H, Guigonis V, Mougenot B, Decobert F, Haymann JP, Bensman A, et al. Antenatal membranous glomerulonephritis due to anti-neutral endopeptidase antibodies. N Engl J Med. 2002;346(26):2053-60. Epub 2002/06/28. doi: 10.1056/NEJMoa012895. PubMed PMID: 12087141.

---

## [Editor Report · Decision Letter 2]

17 Jan 2025

PONE-D-24-36618R2Podocyte specific knockout of the natriuretic peptide clearance receptor is podocyte protective in focal segmental glomerulosclerosisPLOS ONE

Dear Dr. Spurney,

Thank you for submitting your manuscript to PLOS ONE. After careful consideration, we feel that it has merit but does not fully meet PLOS ONE’s publication criteria as it currently stands. Therefore, we invite you to submit a revised version of the manuscript that addresses the last point that should be clarified: Please make sure that the single-cell RNAseq databases you cite (refs 14 and 16) indeed show the expression of NPPC (it can not be found in https://susztaklab.com/EH/par). If you also can't find it, please cite other references or change the text.

We look forward to receiving your revised manuscript.

Kind regards,

Michael Bader

Academic Editor

PLOS ONE
---

## [Author Response · Author response to Decision Letter 2]

22 Jan 2025

RESPONSE TO THE REVIEW

Comments to editorial staff:

I am submitting the revised manuscript entitled "Podocyte specific knockout of the natriuretic peptide clearance receptor ameliorates focal segmental glomerulosclerosis” for consideration for publication in Plos One. I am responding to the following comment:

“Please make sure that the single-cell RNAseq databases you cite (refs 14 and 16) shows expression of NPPC (it cannot be found in https://susztaklab.com/EH/par). If you also can't find it, please cite other references or change the text.”

There are 3 references (14, 16 and 50) with links to the data bases that can be searched to locate the mRNAs by either clinking on the link or copying and pasting the link into a browser. The above link for reference 14 is incorrect (https://susztaklab.com/EH/par).

The links to the correct data sets are:

- Reverence 14: https://susztaklab.com/EH/park/

- Reference 16: http://humphreyslab.com/SingleCell/

- Reference 50: http://humphreyslab.com/SingleCell/

The links in the previous submission were correct and the only changes I have made in the manuscript were to modify the sentence identifying the data bases from, “Results for specific mRNAs can be accessed at the following database” to “Results for specific mRNAs can be accessed by searching the following database”. If this format cannot be used for references, I will eliminate references 14, 16 and 50.

Comments to the authors:

1. If the authors have adequately addressed your comments raised in a previous round of review and you feel that this manuscript is now acceptable for publication, you may indicate that here to bypass the “Comments to the Author” section, enter your conflict of interest statement in the “Confidential to Editor” section, and submit your "Accept" recommendation.

- Reviewer #2: No response

- We have responded to individual comments below.

2. Is the manuscript technically sound, and do the data support the conclusions? The manuscript must describe a technically sound piece of scientific research with data that supports the conclusions. Experiments must have been conducted rigorously, with appropriate controls, replication, and sample sizes. The conclusions must be drawn appropriately based on the data presented.

- Reviewer #2: Partly

- We have made changes in the manuscript as outlined below to address comments raised by the reviewers.

3. Has the statistical analysis been performed appropriately and rigorously?

- Reviewer #2: Yes.

4. Have the authors made all data underlying the findings in their manuscript fully available?

- Reviewer #2: Yes

5. Is the manuscript presented in an intelligible fashion and written in standard English?

- Reviewer #2: Yes

Reviewer comments to the authors:

1. As suggested, experiments were carried out on immortalized podocytes to examine the relevance of CNP signaling via NPRB. For this purpose, the substance P19 was used (Figure 1G). The effect of P19 in KD cells is low and the significance is very limited with a test number of n=4. This could easily be corrected by additional experiments. Furthermore, the concentration of P19 used is not indicated in the manuscript. This is particularly important because the affinity of P19 for NPRA and NPRB does not differ greatly, and the greatest affinity is actually for NPRC. This information can be found in reference 26 and should at least be discussed with all its consequences. A knockdown of NPRB would be the better method here.

We have performed the experiments recommended by the reviewers and, although we only used an “N” of 4, the data is statistically significant, even after correcting for multiple comparisons. We do apologize for not including the concentrations of P19 (250 nM) in the manuscript, and this data has been added to the Methods Section. As the reviewers mentioned, the affinity of P19 for NPRA and NPRB does not differ greatly, and the greatest affinity is actually for NPRC. This is an important consideration that we did not address in the revised manuscript. Based on the following reference (Peptides 26 (2005) 517–524), we suggest that binding of P19 to either NPRC or NPRA is unlikely to affect the results in the knockdown (KD) podocytes because: 1. NPRC is absent in the KD cells, 2. The dosage of P19 used in the studies (250 nM) effectively binds NPRB (Kd 15.4 nM) but binds minimally to NPRA (NPRA Kd 575 nM). In contrast to the KD cells, NPRC is expressed in the control cells, which could affect the apoptotic response by blocking natriuretic clearance and enhancing CNP levels. We posit that blockade of NPRC in the control cells by P19 (Fig. 1G) had a minimal effect on the results because P19 has high affinity for both NPRB (Kd ~15 nM) and NPRC (0.0134 nM) and, in turn, would effectively inhibit both NP receptors (NPRB and NPRC). In this scenario, the benefits of increasing CNP levels by blockade NPRC would be reduced by pharmacologic blockade of NPRB. In the current study, the net effect was no change in the apoptotic response (Fig. 1G).

NOTE: affinity in the P19 reference is expressed as pK units and can be converted to Kd as follows pK = −log KD).

2. The experiments comparing the effects of ANP and CNP on cGMP and apoptosis are very helpful. They clearly show that ANP exerts a protective effect on podocytes even without simultaneous inactivation of NPRC. This fits very well with in vivo experiments by other groups. However, the finding that a protective effect of CNP is only present when NPRC is inactivated calls into question the in vivo relevance of this effect, since podocytes in vivo also express NPRC. This should be discussed.

We agree that, under the conditions used in the experiments (cell culture), ANP is more effect than CNP at inhibiting podocyte apoptosis at equimolar concentrations of ANP and CNP (Figure 2D). Indeed, the podocyte protective actions of CNP are only seen in the NPRC KD cells. The current studies do not, however, address potential CNP secretion under in vivo conditions or changes in CNP secretion during stress. Moreover, there are multiple CNP secreting cells in kidney that may contribute to enhance CNP levels and “overcome” the negative regulatory effects of NPRC. However, based on the data presented in the manuscript, we have modified the statements about the autocrine protective actions of CNP in the text of the manuscript to acknowledge the lack of a podocyte protective effect in the cells expressing NPRC

3. As in the first version of the manuscript, the references should be carefully checked. For example, references 14 and 16 are mentioned to prove that “CNP is widely expressed in multiple cell types in mouse and human kidneys including glomerular podocytes (results, first paragraph)”. This information cannot be found in the papers. There may be a misunderstanding because “CNP” is mentioned in the gene lists. However, this is the gene of a “cyclic nucleotide phoshodiesterase” and not CNP whose gene name is NPPC. In addition, the references are used as evidence for “for the formation of cGMP in podocytes by the NPRB” (results, first paragraph). This is also not correct. Please correct me if I am wrong.

The references 14 and 16 are single cell RNA sequencing studies. To obtain results for specific mRNAs investigators need to query the transcriptomics databases that accompany the manuscripts.

Database for these results, are available at the following websites: 1. http://humphreyslab.com/SingleCell/ and 2. https://susztaklab.com/EH/park/. I have also added these weblinks to each manuscript citing single cell RNA sequencing studies in the Reference Section. I am also requesting guidance how to reference these results in the manuscript.

4. It should be briefly discussed that the blood pressure values show a very high variability. Although this can occasionally occur with automated tail cuff measurement and is therefore not very unusual, it significantly weakens the authors' conclusion that there are no blood pressure differences between the genotypes.

The tail cuff blood pressure measurement did show variability in the results, which reduces chances of detecting a difference between groups. We can only speculate on the causes, but the major finding in the BP studies is the lack of a difference in the Gq mice compared to the Gq-KO mice. This is important because a difference in BP could affect the severity of the kidney disease. The comparison in these 2 groups (GQ and Gq-KO) found very similar BP levels suggesting that a difference in BP did not affect the in vivo findings in the study.

5. Legend Figure 2B: “KO mice” should be corrected to “KD cells”

We have corrected this mistake in the text of the manuscript.

Additional issues: PLOS authors have the option to publish the peer review history of their article (what does this mean?). If published, this will include your full peer review and any attached files.

I do not think is necessary to publish the full review process.

References

1. Kubiak-Wlekly A, Perkowska-Ptasinska A, Olejniczak P, Rochowiak A, Kaczmarek E, Durlik M, et al. The Comparison of the Podocyte Expression of Synaptopodin, CR1 and Neprilysin in Human Glomerulonephritis: Could the Expression of CR1 be Clinically Relevant? Int J Biomed Sci. 2009;5(1):28-36. PubMed PMID: 23675111; PubMed Central PMCID: PMCPMC3614758.

2. Debiec H, Guigonis V, Mougenot B, Decobert F, Haymann JP, Bensman A, et al. Antenatal membranous glomerulonephritis due to anti-neutral endopeptidase antibodies. N Engl J Med. 2002;346(26):2053-60. Epub 2002/06/28. doi: 10.1056/NEJMoa012895. PubMed PMID: 12087141.

---

## [Editor Report · Decision Letter 3]

28 Jan 2025

PONE-D-24-36618R3Podocyte specific knockout of the natriuretic peptide clearance receptor is podocyte protective in focal segmental glomerulosclerosisPLOS ONE

Dear Dr. Spurney,

Thank you for submitting your manuscript to PLOS ONE. After careful consideration, we feel that it has merit but does not fully meet PLOS ONE’s publication criteria as it currently stands.Reviewer 1 and myself had no problem to find and search the single-cell RNAseq databases you cited and we could find the gene CNP to be expressed, but it codes for cyclic nucleotide phoshodiesterase, not for C-type natriuretic peptide. However we did not find NPPC (the gene encoding C-type natriuretic peptide) to be mentioned as gene being expressed in kidney. Therefore, we invite you to submit a revised version of the manuscript that addresses this point by either making sure that these databases contain the information on NPPC expression or citing other supporting references for the renal expression of this gene encoding CNP to support your sentence “CNP is widely expressed in multiple cell types in mouse and human kidneys including glomerular podocytes".

We look forward to receiving your revised manuscript.

Kind regards,

Michael Bader

Academic Editor

PLOS ONE
---

## [Author Response · Author response to Decision Letter 3]

30 Jan 2025

Jnauary30, 2025

To whom it may concern:

I am submitting the revised manuscript entitled "Podocyte specific knockout of the natriuretic peptide clearance receptor ameliorates focal segmental glomerulosclerosis” for consideration for publication in Plos One. I am responding to the following comment:

“Reviewer 1 and myself had no problem to find and search the single-cell RNAseq databases you cited, and we could find the gene CNP to be expressed, but it codes for cyclic nucleotide phosphodiesterase, not for C-type natriuretic peptide. However, we did not find NPPC (the gene encoding C-type natriuretic peptide) to be mentioned as gene being expressed in kidney.”

We have retained the citations in the manuscript relevant to CNP expression in the kidney, but we have removed references to CNP expression in then following data bases:

- Reverence 14: https://susztaklab.com/EH/park/

- Reference 16: http://humphreyslab.com/SingleCell/

- Reference 50: http://humphreyslab.com/SingleCell/

These data bases are relevant to expression of natriuretic peptide receptors (NPRA, NPRB and NPRC) and these references remain in the manuscript citations. Changes to text of the manuscript have been highlighted in red.

Thanks for the careful review.

Correspondence regarding the manuscript should be sent to the following address:

Robert F. Spurney, M.D.

MSRB 2, Room 2013

106 Research Drive

Durham, N.C. 27710

(919)-684-9729

FAX: (919)-684-3011

E-mail: robert.spurney@duke.edu

---

## [Editor Report · Decision Letter 4]

2 Feb 2025

Podocyte specific knockout of the natriuretic peptide clearance receptor is podocyte protective in focal segmental glomerulosclerosis

PONE-D-24-36618R4

Dear Dr. Spurney,

We’re pleased to inform you that your manuscript has been judged scientifically suitable for publication and will be formally accepted for publication once it meets all outstanding technical requirements.

Kind regards,

Michael Bader

Academic Editor

PLOS ONE
---

## [Editor Report · Acceptance letter]

PONE-D-24-36618R4

PLOS ONE

Dear Dr. Spurney,

I'm pleased to inform you that your manuscript has been deemed suitable for publication in PLOS ONE. Congratulations! Your manuscript is now being handed over to our production team.

Kind regards,

on behalf of

Prof. Michael Bader

Academic Editor

PLOS ONE